# SimVAE: Narrowing the gap between Discriminative & Generative Representation Learning

## Abstract

Self-supervised learning (SSL) is a powerful representation learning paradigm that leverages an auxiliary task, such as identifying different augmentations, extracts, or modalities of a common sample. Recent methods, e.g. SimCLR, CLIP and DINO, have made significant strides, yielding representations that achieve state-of-the-art results on multiple downstream tasks. A number of *discriminative* SSL approaches have been proposed, e.g. instance discrimination, latent clustering and contrastive learning. Although they often seem intuitive, a theoretical understanding of their underlying mechanisms eludes. Meanwhile, representations learned by *generative* methods, such as variational autoencoders (VAEs), fit a specified latent variable model and have principled appeal, but lag significantly in terms of performance. We analyse several discriminative self-supervised methods and propose a graphical model to reflect the assumptions they implicitly make in latent space, providing a unifying theoretical framework for these methods. We show that fitting this model (*SimVAE*) to the data improves representations over other VAE-based methods on several common benchmarks (MNIST, FashionMNIST, CIFAR10, Celeb-A), narrows the gap to discriminative methods on standard classification tasks and even *outperforms* them on task that require more *stylistic* information to be captured.

## 1 Introduction

Self-supervised learning (SSL) has become a prominent approach to unsupervised representation learning. Under this paradigm, a model is trained to perform an auxiliary task without annotated labels, such that representations of the data are learned in the process that generalize to a range of downstream tasks. Recently, contrastive SSL approaches have achieved remarkable performance and garnered significant attention, exemplified by algorithms such as InfoNCE (Oord et al., 2018), SimCLR (Chen et al., 2020), SWaV (Caron et al., 2020) and CLIP (Radford et al., 2021). These methods exploit semantically related observations, such as different parts (Oord et al., 2018; Mikolov et al., 2013), augmentations (Chen et al., 2020; Misra & Maaten, 2020), or modalities/views (Baevski et al., 2020; Radford et al., 2021; Arandjelovic & Zisserman, 2017) of the data, that share latent *content* information and differ in *style*. A variety of SSL methods has arisen based on a range of intuitive but heuristic strategies and design choices (Balestriero et al., 2023). Previous works have analysed SSL methods (Wang & Isola, 2020; Zimmermann et al., 2021; Tian, 2022), but a principled, mathematical mechanism justifying their performance remains unclear, limiting confidence in their reliability, their interpretability and the development of improved algorithms. Meanwhile, representations can be learned from more principled approaches, such as using variational inference to learn parameters of a latent variable model (Kingma & Welling, 2014), but tend to underperform SSL representations.

To address this, we consider the relationship between *discriminative* and *generative* representation learning and the correspondence between an *encoder* $f : \mathcal{X} \to \mathcal{Z}$, that maps data samples $x \in \mathcal{X}$ to representations $z \in \mathcal{Z}$, and the posterior $p(z|x)$ under a generative model. From this perspective, we analyse the implicit latent structure induced by several discriminative self-supervised algorithms, including the popular InfoNCE objective adopted by numerous SSL models (e.g. Chen et al., 2020; Radford et al., 2021). We show that those methods reflect a *common* hierarchical latent variable model for the data (Figure 2), under which semantically related samples are generated from the same cluster in the latent space, giving them common *content*. We also show that discriminative losses may "collapse" these clusters, potentially losing distinguishing *style* information that a downstream task may require (e.g., object location). This effectively prioritises some downstream tasks over others,

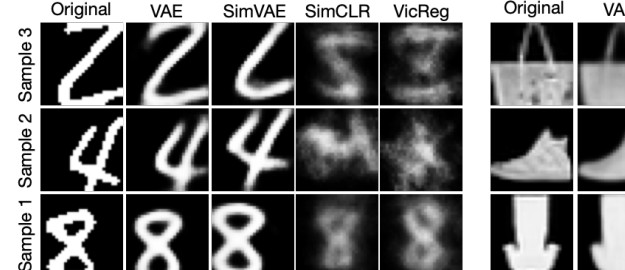

Figure 1: Qualitative assessment of representation information: images reconstructed from representations learned by generative unsupervised learning (VAE), generative SSL (our SimVAE) and discriminative SSL (SimCLR, VicReg). Datasets: MNIST (*l*), Fashion MNIST (*r*), original images in left columns. Style information (i.e., position & orientation) is lost through discriminative learning.

limiting the generality of representations. We therefore propose *SimVAE*, a *generative* approach to self-supervised learning that explicitly models the latent structure we identify as implicit in discriminative methods.[1] We derive the associated evidence lower bound (ELBO) as the SimVAE training objective, providing a principled approach to self-supervised representation learning. Notably, where discriminative methods propose a variety of ways to avoid representations "collapsing", the reconstruction aspect of SimVAE prevents that automatically since to reconstruct distinct samples requires their representations to be distinct.

Generative methods are, however, generally more challenging to train than their discriminative counterparts, not least because probability distributions must be well modelled, hence we do not expect to immediately bridge the performance gap between these paradigms. Encouragingly though, our results show that SimVAE is competitive with, and even outperforms, popular discriminative methods on downstream classification accuracy on simple datasets (MNIST, FashionMNIST), suggesting that SimVAE is a promising approach if distributions can be well modelled. On more complex datasets (CIFAR10, Celeb-A), SimVAE is less competitive for content classification, but consistently outperforms discriminative methods on tasks requiring *stylistic* information. On all tasks, SimVAE significantly outperforms (>15%) other VAE-based generative models.

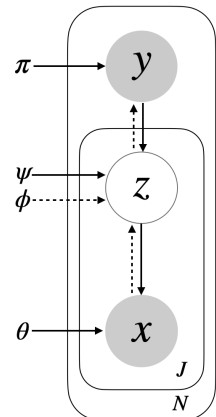

Figure 2: Graphical model for SSL with *J* related samples.

We believe that these results justify further research in this challenging direction given the benefits that generative representation learning offers, such as: a measure of uncertainty given by the posterior, the ability to generate synthetic samples as a bi-product, and the ability to qualitatively assess the information captured in representations by inspecting their regenerations (Figure 1). Perhaps most importantly for learning general purpose representations, rather than learning *invariant* representations that *lose* information (typically *style*, e.g. object location), a generative approach aims to preserve information, with even the prospect of *disentangling* it (Higgins et al., 2017). Our generative SimVAE model also connects SSL to latent variable models of other learning paradigms, such as fully unsupervised learning (VAEs) and supervised learning (Variational Classification, Dhuliawala et al., 2023), which may enable them to be combined in a principled and unified learning regime. Our main contributions are as follows:

- we provide theoretical insight into the representations learned by a number of popular self-supervised learning algorithms and propose a unifying latent variable model (Figure 2);
- we introduce SimVAE, a generative approach to self-supervised representation learning inspired by our analysis of discriminative methods (§4.2);
- we demonstrate SimVAE's performance at prediction tasks on standard datasets, showing a clear improvement (e.g., +15% on CIFAR10) over previous VAE-based representations, including one adapted to SSL (Sinha & Dieng, 2021) (§6);
- we show qualitatively (Figure 1) and quantitatively (Figure 3) that SimVAE captures more style information relative to several discriminative methods.

---

[1]So-named because it can be viewed as encoding the latent structure of **Sim**CLR (Chen et al., 2020) in the prior of a variational auto-encoder (**VAE**) (Kingma & Welling, 2014).

## 2 BACKGROUND AND RELATED WORK

**Representation Learning**: We consider methods where an *encoder* function $f : \mathcal{X} \to \mathcal{Z}$ learns to map data $x \in \mathcal{X}$ to a (typically lower-dimensional) representation $z = f(x) \in \mathcal{Z}$ that should perform well in place of $x$ in downstream tasks. Representation learning is not well defined since the downstream task can be arbitrary and a representation that performs well on one may perform poorly on another (Zhang et al., 2022). For instance, unsupervised image representations are commonly evaluated by predicting unseen class labels, but a downstream task might be to detect lighting, position or orientation, which a representation good at class prediction may not capture. This suggests that *general* unsupervised representations should capture as much information about the data as possible. Recent works support this by assessing an variety of downstream tasks (Balažević et al., 2023).

**Self-Supervised Learning (SSL)** leverages unsupervised auxiliary tasks to learn representations and spans many approaches (Balestriero et al., 2023). We categorise the classes we address below.

*Instance Discrimination* (e.g. Dosovitskiy et al., 2014; Wu et al., 2018) treats each data sample $x_i$, and any of its augmentations, as a distinct class $i$ and trains a softmax classifier to predict the "class", with encoder outputs used as representations.

*Contrastive Learning* encourages representations of *semantically related data* (positive samples) to be "close" relative to pairs sampled at random (negative samples). Some early SSL approaches use an energy-based loss (Chopra et al., 2005; Hadsell et al., 2006). The *word2vec* algorithm (W2V, Mikolov et al., 2013) probabilistically predicts words that co-occur over a text corpus, and its embeddings learn to factorise the *pointwise mutual information* (PMI) between words (Levy & Goldberg, 2014), known to cause semantic properties in the embeddings (Allen & Hospedales, 2019). InfoNCE (Sohn, 2016; Oord et al., 2018) adapts W2V to other domains, e.g. $x, x'$ can be image patches or sound clips. The InfoNCE loss for $x$, positive sample $x^+$ and negative samples $X^- = \{x_l^-\}_{l=1}^L$ is defined:

$$\mathcal{L}_{NCE}(x, x^+, X^-) = \log \frac{\exp \text{sim}(z, z^+)}{\exp \text{sim}(z, z^+) + \sum_{x^- \in X^-} \exp \text{sim}(z, z^-)}, \qquad (1)$$

where $z$ is the representation of $x$ and $\text{sim}(\cdot, \cdot)$ is a *similarity function*, e.g. dot product. Equation 1 is minimised if $\text{sim}(z, z') = \text{PMI}(x, x') + c$, where $c$ is a constant (that can vary with $x$) (Oord et al., 2018). Many works build on InfoNCE, e.g. SimCLR (Chen et al., 2020) uses synthetic augmentations and CLIP (Radford et al., 2021) other modalities as positive samples; DIM (Hjelm et al., 2019) uses different encoder parameters as representations; and MoCo (He et al., 2020), BYOL (Grill et al., 2020) and VicReg (Bardes et al., 2022) propose novel strategies to avoid negative sampling.

*Latent Clustering* methods cluster representations during training (for an assumed cluster number), which both identifies clusters and encourages them to form. Early examples apply K-means or similar clustering to the hidden layer of a deterministic auto-encoder (Song et al., 2013; Xie et al., 2016; Yang et al., 2017). DeepCluster (Caron et al., 2020) iteratively clusters ResNet representations by K-means, and predicts cluster assignments as "pseudo-labels". DINO (Caron et al., 2021), a transformer-based model, can be interpreted as performing clustering in the latent space (Balestriero et al., 2023).

*Other SSL approaches* (that we do not address) include methods that reconstruct data samples from perturbations, such as randomly masked patches (He et al., 2022; Xie et al., 2022); and methods that predict details of an applied perturbation, such as rotation angle (Gidaris et al., 2018).

**Variational Auto-Encoder (VAE)**: Assuming a latent generative model $Z \to X$, parameters $\theta$ of a model $p_\theta(x) = \int_z p_\theta(x|z)p_\theta(z)$ can be learned by maximising the *evidence lower bound* (**ELBO**):

$$\int_x p(x) \log p_\theta(x) \;\; \geq \;\; \int_x p(x) \int_z q_\phi(z|x) \log \frac{p_\theta(x|z)}{q_\phi(z|x)} + \log p(z) \;\; \doteq \;\; \text{ELBO} \qquad (2)$$

By maximising Eq. 2, the model approximates the data distribution $p(x)$, and $q_\phi(z|x)$ approximates the (typically intractable) posterior $p_\theta(z|x)$. A VAE (Kingma & Welling, 2014) implements the ELBO as a training objective with $p_\theta$ and $q_\phi$ modelled as Gaussians parameterised by neural networks. Latent variables provide a natural representation of the data (discussed further in §3). The $\beta$-VAE (Higgins et al., 2017) weights the entropy and prior terms of Eq. 2 to improve *disentanglement* of latent dimensions. More relevant to SSL, CR-VAE (Sinha & Dieng, 2021) minimises a KL divergence between posteriors of semantically related samples to Eq. 2 to bring their latent variables, or representations, together.

VAEs have been extended in many ways to model more complex generative processes. In particular, hierarchical VAEs (e.g. Valpola, 2015; Ranganath et al., 2016; Rolfe, 2017; He et al., 2018; Sønderby et al., 2016; Edwards & Storkey, 2016) model hierarchical latent variable structure, which our

SimVAE model falls under. In particular, Edwards & Storkey (2016) propose a graphical model similar to Figure 2 where a context variable $c$ applies to a data *set* (equivalent to our $y$). A notable difference is in how the posteriors are factorised, which is critical to allowing representations to be inferred independently of related data. Wu et al. (2023) and Sinha et al. (2021) combine loss components from VAEs and contrastive learning but do not propose a general hierarchical latent variable model for self-supervised learning; and Nakamura et al. (2023) look to explain various SSL methods via the ELBO, but using a second model of the posterior distribution rather than a generative model.

**Variational Classification (VC)**: Dhuliawala et al. (2023) present a latent variable model for classification, $p(y|x) = \int_z q(z|x) \frac{p(z|y)p(y)}{p(z)}$ for labels $y$, generalising softmax neural network classifiers and interpreting them as an *encoder* $f : \mathcal{X} \to \mathcal{Z}$ parameterising $q(z|x)$; and a softmax layer encoding $p(y|z)$ by Bayes' rule. For a softmax classifier, the optimal $q^*(z|x)$ for all $x$ of a given class are shown to be a *common delta distribution*, meaning that representations of a class are *collapsed* together, despite the distribution $p(z|y)$ encoded in the softmax layer (in practice, constraints such as $l_2$ regularisation arbitrarily restrict the optimum being reached). Any discrepancy between the distribution that representations *follow* "$q(z|y)$" vs that *anticipated* $p(z|y)$ is seen to have negative consequences, e.g. miscalibration, and is relevant to SSL methods involving softmax (e.g. instance discrimination).

Several works look to provide a latent interpretation of SSL (Tian, 2022; Nakamura et al., 2023; Sansone & Manhaeve, 2022), however none proposes a unifying latent variable model (Fig. 2).

## 3 Contrastive vs Generative Representation Learning

Representation learning approaches can be discriminative or generative. Many recent self-supervised approaches are discriminative and train an encoder under a loss function that induces intuitively desirable properties in the representation space $\mathcal{Z}$, e.g. representations of semantically related data samples being "close" relative to random samples. Meanwhile, a generative latent variable model $p(x) = \int_z p(x|z)p(z)$ can be interpreted as first sampling a latent variable $z \sim p(\mathrm{z})$ that defines the underlying characteristics of a data point; then sampling $x \sim p(\mathrm{x}|z)$ to obtain a manifestation of those properties. Hence, the posterior $p(z|x)$, which effectively *reverses the generative process* to infer $z$, and so the semantic properties of $x$, naturally provides a semantically meaningful representation of $x$. Since we often require a single succinct representation, *cf* dimensionality reduction, the *mean* of $p(z|x)$, often parameterised by an encoder, is a natural choice and the distribution can be interpreted as a measure of uncertainty (more generally the posterior could be used in a fully Bayesian manner).

Discriminative and generative paradigms are, in fact, closely related since a discriminatively trained encoder $f : \mathcal{X} \to \mathcal{Z}$ can be interpreted as a posterior (delta-)distribution $p_f(z|x) \doteq \delta_{z-f(x)}$. Together with the data distribution $p(x)$, this gives a joint distribution that implicitly defines a latent distribution $p_f(z) \doteq \int_x p_f(z|x)p(x)$, and likelihood $p_f(x|z)$ by Bayes' rule. Hence *spatial* properties imposed on representations by a discriminative loss function implicitly impose *distributional* properties in $p_f(z)$. Thus, fitting a latent variable model to the data with a prior $p^*(z)$ might be attempted: generatively, by maximising the respective ELBO (Eq. 2); or discriminatively, by designing a loss function that imposes spatial constraints on representations such that the (optimised) encoder induces the chosen prior, i.e. $p_f(z) = p^*(z)$. Our central premise is that various deterministic SSL methods induce a similar distributional structure in the latent space, which we identify and approach generatively.

In practice, training a generative model under the ELBO is often more challenging than optimising a discriminative loss as $p(x|z)$ must also be learned and the ELBO may have multiple solutions, particularly when using flexible distributions parameterised by neural networks, whereby latents $z$ are only identifiable up to certain symmetries, e.g. rotation (Khemakhem et al., 2020; Locatello et al., 2019). On the other hand, generative approaches offer a principled basis for representation learning, an uncertainty estimate over representations (i.e. the posterior), the ability to generate synthetic data, and qualitative insight into the features representations capture by considering their regenerations.

Given the interplay between generative and discriminative SSL, we investigate the latent structure induced by several discriminative methods, including InfoNCE (Oord et al., 2018), which underpins popular methods such as SimCLR and CLIP (§4). We posit a latent variable model to formalise the latent structure we intuit that those methods induce, and derive the respective ELBO as a principled

training objective to fit that model to the data. Lastly, we train representations under this generative approach, named *SimVAE*, and compare their performance on downstream tasks (§5).

# 4 SELF-SUPERVISED REPRESENTATION LEARNING

We first consider what it means for a subset of data $X_i = \{x_i^j\}_{j=1}^J \subset \mathcal{X}$ to be *semantically related*. We interpret this to imply a relationship between latent variables $\{z_i^j\}_{j=1}^J$ that respectively define properties of $x_i^j \in X_i$. Specifically, we assume $z_i^j \sim p(\mathrm{z}|y)$ are independent samples from a common distribution conditioned on another latent random variable $y$ (continuous or discrete), distributed $p(\mathrm{y})$, where $y$ indexes the latent property that $x_i^j$ share, termed *semantic content*. To motivate this, note that if conditionals $p(\mathrm{z}|y)$ are concentrated relative to the variance of $p(\mathrm{z})$, representations of semantically related data are *clustered*, mirroring contrastive methods that "pull together" representations of related samples (§2). For a dataset $X = \bigcup_{i=1}^N X_i$, of $N$ subsets of semantically related data, e.g. augmentations or extracts from larger data samples, this latent structure can be formalised by the hierarchical latent variable model in Figure 2. The marginal probability and ELBO for a single data sample is given by (for continuous $y$, the discrete case is analogous):

$$p_\theta(x) \; = \; \int_{z,y} p_\theta(x|z) p_\theta(z|y) p_\theta(y) \tag{3}$$

$$\log p_\theta(x) \; \geq \; \int_z q_\phi(z|x) \log \tfrac{p_\theta(x|z)}{q_\phi(z|x)} + \int_z q_\phi(z|x) \int_y q_\phi(y|z) \log \tfrac{p_\theta(z|y) p_\theta(y)}{q_\phi(y|z)} \tag{4}$$

Note that equation 4 extends the standard ELBO (Equation 2) used to train a VAE by replacing the log prior with a lower bound: $\log p_\theta(z) \geq \int_y q(y|z) \log \tfrac{p_\theta(z|y) p(y)}{q(y|z)}$. Note also that the semantic content represented by $y$ may be instance-level (a specific dog), class-level (any dog), in between (a particular dog breed) or beyond (an animal). If $y$ is discrete, the prior $p_\theta(z) = \sum_y p_\theta(z|y) p_\theta(y)$ is a mixture model and the exact posterior $p_\theta(y|z)$ can be computed analytically by Bayes' rule. The EM algorithm can then be used to fit parameters of a discrete mixture model, e.g. by K-means, hence optimising Equation 4 for discrete $y$, is comparable to training an auto-encoder while clustering the latent space. Methods taking this approach (Song et al., 2013; Xie et al., 2016; Yang et al., 2017) can therefore be considered to approximate the latent structure in Figure 2 (for $J = 1$). Note that the model so far is fully unsupervised, we next consider self-supervised methods.

## 4.1 DISCRIMINATIVE SELF-SUPERVISED LEARNING

Here, we consider several approaches to discriminative self-supervised learning, how they relate to one another and the generative latent variable model in Figure 2.

**Instance Discrimination (ID)** (Dosovitskiy et al., 2014; Wu et al., 2018) trains a classifier using the index $i \in [1, N]$ as the "label" for each sample $x_i$ and its augmentations (if any). From *Variational Classification* (VC, Dhuliawala et al., 2023, §2) the softmax cross-entropy loss can be viewed from a latent perspective under the latent variable model i → z → x (*cf* Figure 2):

$$\log p(i|x_i^j) \geq \int_z q_\phi(z|x_i^j) \log p(i|z) \tag{5}$$

where, $j$ indexes augmentations (including identity). VC shows that by maximising Equation 5 for mutually exclusive classes (as samples and their augmentations are typically assumed to be), representations of each "class" $i$, defined by optimal distributions $q_\phi(z|x_i^j)$, "collapse" together and distinct classes spread apart. Since classes are defined as samples varying in *style*, ID learns representations invariant to those differences ("transformation invariant", Dosovitskiy et al., 2014).

**Deep Clustering (DC)** (Caron et al., 2020) repeatedly performs K-means clustering on representations from a ResNet encoder (for a large fixed number of clusters), which is found to cluster semantically related samples due to the inductive bias of a ResNet. Current cluster assignments are used as pseudo-labels to train a softmax layer with constraints to balance class assignments. While semantically related samples are defined differently, this approach induces clustering as in ID, and the use of a softmax classifier will again collapse together representations in a cluster.

**Contrastive Learning** is exemplified by the InfoNCE objective (Eq 1), equivalent to instance discrimination by softmax classification but normalising over a mini-batch rather than all "classes"

(instances); and having "non-parametric" similarity functions of embeddings rather than class-specific parameters (Wu et al., 2018). The InfoNCE objective is known to be optimised if embeddings can satisfy $sim(z, z') = \text{PMI}(x, x') + c$ (§2). However, using the popular *bounded* cosine similarity $cossim(z, z') = \frac{z^\top z'}{\|z\|\|z'\|} \in [-1, 1]$ (e.g. Chen et al., 2020) means that optimal embeddings of semantically related data (PMI $> 0$) have the same orientation, $cossim(z, z') = 1$, and otherwise (PMI$=-\infty$) be spread apart, $cossim(z, z') = -1$.[2] (for a fuller discussion see Appendix A.1)

In summary, we consider discriminative SSL methods with a range of different loss functions, spanning instance discrimination, latent clustering and contrastive learning, and find that they all cluster representation of semantically related data. This suggests that the induced latent distributions $p_f(z)$ can be formalised by the common hierarchical latent variable model in Figure 2, where $y$ reflects common semantic *content* and representations $z|y$ of semantically related data, i.e. conditioned on the same $y$, are clustered together. Furthermore, semantically related data are not simply clustered in these methods, but are *collapsed* together, losing information that differentiates them, that might be important for a downstream task. This, in effect, gives primacy to *content* over *style* and **prioritises certain downstream tasks over others**, which seems antithetical to general representation learning. Indeed, style information is important for many real-world tasks, e.g. detecting facial expression, voice sentiment, or the surroundings of an RL agent; and is of central importance in other branches of representation learning (Higgins et al., 2017; Karras et al., 2019). In essence, by preserving *content* at the expense of other information, **contrastive methods may *over-fit* representation learning to style-agnostic tasks**. In supervised settings, discriminatively learned representations also give over-confident, or miscalibrated predictions (Dhuliawala et al., 2023) because, as well as style information, the relative probability of different samples is discarded. This information will similarly be lost in discriminative SSL methods, despite again being of potential use in downstream tasks. To try to avoid these pitfalls, we consider a generative SSL approach to learning representations under the identified latent variable model.

## 4.2 Generative Self-Supervised Learning (SimVAE)

Having intuited that the various discriminative SSL methods considered in §4.1 induce comparable latent structure described by the graphical model in Figure 2, we now consider a generative approach to achieving the same. For $J$ *semantically related* samples $\mathbf{x} = \{x^j\}$ we have:

$$p(\mathbf{x}) = \int_{\mathbf{z}, y} \Big( \prod_j p(x^j|z^j) \Big) \Big( \prod_j p(z^j|y) \Big) p(y) \tag{6}$$

$$\log p(\mathbf{x}) \geq \sum_j \int_{z^j} q(z^j|x^j) \log \frac{p(x^j|z^j)}{q(z^j|x^j)} + \int_{\mathbf{z}} q(\mathbf{z}|\mathbf{x}) \int_y q(y|\mathbf{z}) \Big[ \sum_j \log \frac{p(z^j|y)}{q(y|z^j)} + \log p(y) \Big] \tag{7}$$

where $\mathbf{z} = \{z^j\}$ and the approximate posterior is assumed to factorise $q(\mathbf{z}|\mathbf{x}) \approx \prod q(z^j|x^j)$.[3] A derivation of Equation 7 is given in Appendix A.2. In simple terms, the proposed *SimVAE* model can be considered a VAE with a mixture model prior $p(\mathbf{z}) = \int_y p(\mathbf{z}|y)p(y)$, and semantically related samples are conditioned on the same $y$. While other choices can be made, we assume all $p(\mathbf{z}|y)$ are Gaussian with $y$-dependent mean and common fixed variance $\sigma^2$ that is low relative to the variance of $p(z)$. Hence representations of semantically related samples are "pinched together" in latent space. As in Eqs. 3 and 4, $y$ can be discrete or continuous, e.g. DeepCluster assumes a finite number of discrete clusters. If $y$ is discrete, each $p(z|y)$ must be parameterised, which can be memory intensive for a large number of clusters; and computing the sum over all values of $y$ is computationally expensive unless hard cluster assignment is assumed, $q(y|\mathbf{z}_i) = \delta_{y-i}$, as in K-means. If $y$ is continuous, there are effectively an infinite number of clusters, but if $p(y)$ is assumed Gaussian or uniform (over a suitable region) $y$ can be analytically integrated out efficiently. Unlike most discriminative methods, that contain *pairwise* similarity functions of representations, e.g. cosine similarity, Eq. 7 accommodates any number of related samples $J$ to be processed efficiently. Algorithm 1 summarizes the computational steps required to optimise Equation 7, for continuous $y$ and $p(y)$ uniform.

---

[2]We note that Wang & Isola (2020) suggest a similar conclusion but less comprehensively since components of the InfoNCE objective were considered independently and the relationship to PMI not considered.

[3]Expected to be a reasonable assumption for $z^j$ that carry high information w.r.t. $x^j$, such that observing a related $x^k$ or its representation $z^k$ provides negligible extra information, i.e. $p(z^j|x^j, x^k) \approx p(z^j|x^j)$.

---

**Algorithm 1** SimVAE

---

**Require:** data $\{\mathbf{x}_k\}_{k=1}^M$; batch size $N$; data dimension $D$; augmentation set $\mathcal{T}$; latent dimension $L$; number of augmentations $J$; encoder $f_\phi$; decoder $g_\theta$; prior $p(\mathbf{z}|y)$ variance, $\boldsymbol{\sigma}^2$;

    **for** randomly sampled mini-batch $\{\mathbf{x}_k\}_{k=1}^N$ **do**

        $\{t^j\}_{j=1}^J \sim \mathcal{T}$;                                            # augment mini-batch

        $\{\mathbf{x}_k^j\}_{j=1}^J = \{t^j(\mathbf{x}_k)\}_{j=1}^J$;

        $\{(\boldsymbol{\mu}_k^j, \boldsymbol{\Sigma}_k^j) = f_\phi(\mathbf{x}_k^j)\}_{j=1}^J$;               # forward pass : $\mathbf{z} \sim p(\mathbf{z}|\mathbf{x}), \tilde{\mathbf{x}} \sim p(\mathbf{x}|\mathbf{z})$

        $\{\mathbf{z}_k^j \sim \mathcal{N}(\boldsymbol{\mu}_k^j, \boldsymbol{\Sigma}_k^j)\}_{j=1}^J$;

        $\{\tilde{\mathbf{x}}_k^j = g_\theta(\mathbf{z}_k^j)\}_{j=1}^J$;

        $\mathcal{L}_{\text{rec}}^k = \frac{1}{D} \sum_{j=1}^J ||\mathbf{x}_k^j - \tilde{\mathbf{x}}_k^j||_2^2$               # compute & minimize loss terms

        $\mathcal{L}_{\text{H}}^k = \frac{1}{2} \sum_{j=1}^J \log(|\boldsymbol{\Sigma}_k^j|)$

        $\boldsymbol{\mu}_k^* = \frac{1}{J} \sum_{j=1}^J \mathbf{z}_k^j$

        $\mathcal{L}_{\text{prior}}^k = \frac{1}{2} \sum_{j=1}^J ||(\mathbf{z}_k^j - \boldsymbol{\mu}_k^*)/\boldsymbol{\sigma}||_2^2$

        $\min(\mathcal{L} = \frac{1}{N} \sum_{k=1}^N \mathcal{L}_{\text{rec}}^k + \mathcal{L}_{\text{H}}^k + \mathcal{L}_{\text{prior}}^k)$ w.r.t. $\phi, \theta$ by SGD;

    **end for**

    **return** $\phi, \theta$;

---

## 5 EXPERIMENTAL SETUP

**Datasets and Evaluation Metrics** We evaluated SimVAE on four benchmark datasets including two with natural images: MNIST (LeCun, 1998), FashionMNIST (Xiao et al., 2017), Celeb-A (Liu et al., 2015) and CIFAR10 (Krizhevsky et al., 2009). We augment images following the SimCLR (Chen et al., 2020) protocol which includes cropping and flipping as well as color jitter for natural images. We evaluate representations' utility for downstream classification tasks using a linear probe, a non-linear MLP probe, and k-nearest neighbors (kNN) (Cover & Hart, 1967) trained on the pre-trained frozen representations using image labels (Chen et al., 2020; Caron et al., 2020). Additionally, we conducted a fully unsupervised evaluation by fitting a Gaussian mixture model (GMM) to the frozen features for which the number of clusters was set to its ground-truth value. Downstream performance is measured in terms of classification accuracy (CA). A model's generative quality was evaluated using the FID score (Heusel et al., 2017) and reconstruction error (see appendix A.4). For further experimental details and clustering scores please see appendices A.3.1, A.3.2, A.3.5 and A.4.

**Baselines methods** We compare SimVAE to other VAE-based models including the vanilla VAE (Kingma & Welling, 2014), $\beta$-VAE (Higgins et al., 2017) and CR-VAE (Sinha & Dieng, 2021), as well as to state-of-the-art self-supervised discriminative methods including SimCLR (Chen et al., 2020), VicREG (Bardes et al., 2022), and MoCo (He et al., 2020). As a lower bound, we also provide results obtained for randomly initialized embeddings. To ensure fair comparison, the augmentation strategy, representation dimensionality, batch size, and encoder-decoder architectures were kept invariant across methods. To enable a qualitative comparison of representations, decoder networks were trained for each discriminative baseline on top of frozen representations using the reconstruction error. See appendices A.3.3 and A.3.4 for further details on training baselines and decoder models.

**Implementation Details** We use MLP and Resnet18 (He et al., 2016) network architectures for simple and natural image datasets respectively. We fix the dimension of representations $z$ to 10 for MNIST, FashionMNIST, and to 64 for Celeb-A and CIFAR10 datasets. For all generative approaches, we adopt Gaussian posteriors, $q(z|x)$, priors, $p(z)$, and likelihoods, $p(x|z)$, employing diagonal covariance matrices (Kingma & Welling, 2014). For SimVAE, we adopt Gaussian likelihoods $p(z|y)$, $q(y|\mathbf{z}_i) = \delta_{y-i}$ and consider $y$ to be continuous and uniformly distributed. The covariances of the likelihood distributions are tuned and fixed. SimVAE conveniently allows for the simultaneous incorporation of sets of related observations. After tuning, we fix the number of augmentations to 6 and 2 (see Figure 6 for an ablation) for simple and natural datasets respectively. For baselines, all sensitive hyperparameters were tuned independently for each dataset and method. Further details regarding hyperparameters can be found in appendices A.3.3 and A.3.4.

| | | LP-CA | MP-CA | KNN-CA | GMM-CA |
|---|---|---|---|---|---|
| **MNIST** | Random | $39.7 \pm 2.4$ | $38.1 \pm 3.8$ | $46.1 \pm 2.5$ | $42.2 \pm 1.2$ |
| | SimCLR | $\mathbf{98.6} \pm 0.1$ | $\mathbf{98.7} \pm 0.0$ | $\mathbf{98.8} \pm 0.1$ | $\mathbf{97.2} \pm 0.6$ |
| | VicReg | $96.5 \pm 0.2$ | $96.6 \pm 0.1$ | $96.7 \pm 0.1$ | $80.0 \pm 2.3$ |
| | MoCo | $88.6 \pm 1.7$ | $94.6 \pm 0.4$ | $94.6 \pm 0.3$ | $70.5 \pm 4.0$ |
| | VAE | $97.2 \pm 0.2$ | $\mathbf{97.8} \pm 0.1$ | $98.0 \pm 0.1$ | $96.3 \pm 0.4$ |
| | $\beta$-VAE ($\beta = 1.2$) | $97.4 \pm 0.1$ | $\mathbf{97.8} \pm 0.0$ | $98.0 \pm 0.1$ | $96.3 \pm 0.6$ |
| | CR-VAE | $96.6 \pm 0.1$ | $97.2 \pm 0.2$ | $97.6 \pm 0.0$ | $81.3 \pm 2.2$ |
| | SimVAE | $\mathbf{97.6} \pm 0.1$ | $97.9 \pm 0.1$ | $97.8 \pm 0.1$ | $\mathbf{97.0} \pm 0.0$ |
| **Fashion** | Random | $51.2 \pm 0.6$ | $49.8 \pm 0.8$ | $66.5 \pm 0.4$ | $48.6 \pm 0.2$ |
| | SimCLR | $\mathbf{77.4} \pm 0.2$ | $\mathbf{79.0} \pm 0.1$ | $\mathbf{79.3} \pm 0.1$ | $\mathbf{63.6} \pm 2.2$ |
| | VicReg | $70.7 \pm 0.9$ | $72.6 \pm 0.6$ | $76.0 \pm 0.2$ | $57.7 \pm 0.8$ |
| | MoCo | $65.0 \pm 1.3$ | $71.2 \pm 0.1$ | $76.9 \pm 0.2$ | $56.6 \pm 1.1$ |
| | VAE | $77.0 \pm 0.5$ | $80.2 \pm 0.3$ | $83.7 \pm 0.2$ | $57.9 \pm 0.8$ |
| | $\beta$-VAE ($\beta = 1.2$) | $77.2 \pm 0.1$ | $79.7 \pm 0.2$ | $83.5 \pm 0.4$ | $57.5 \pm 0.2$ |
| | CR-VAE | $77.7 \pm 0.4$ | $80.1 \pm 0.1$ | $\mathbf{84.0} \pm 0.2$ | $67.5 \pm 1.2$ |
| | SimVAE | $\mathbf{78.6} \pm 0.0$ | $\mathbf{81.1} \pm 0.1$ | $\mathbf{84.0} \pm 0.0$ | $\mathbf{69.9} \pm 0.0$ |
| **Celeb-A** | Random | $64.4 \pm 0.9$ | $65.3 \pm 1.0$ | $62.0 \pm 0.9$ | $59.2 \pm 0.3$ |
| | SimCLR | $94.2 \pm 0.2$ | $92.7 \pm 0.4$ | $92.0 \pm 0.3$ | $\mathbf{71.6} \pm 0.6$ |
| | VicReg | $\mathbf{94.3} \pm 0.3$ | $\mathbf{94.7} \pm 0.1$ | $\mathbf{92.7} \pm 0.4$ | $53.9 \pm 0.2$ |
| | VAE | $81.5 \pm 1.0$ | $87.7 \pm 0.5$ | $79.6 \pm 0.7$ | $\mathbf{58.8} \pm 0.2$ |
| | $\beta$-VAE ($\beta = 1.2$) | $81.9 \pm 0.2$ | $86.7 \pm 0.4$ | $79.8 \pm 0.1$ | $\mathbf{59.5} \pm 0.6$ |
| | CR-VAE | $81.6 \pm 0.3$ | $87.7 \pm 0.4$ | $79.6 \pm 0.6$ | $\mathbf{58.9} \pm 0.4$ |
| | SimVAE | $\mathbf{87.1} \pm 0.3$ | $\mathbf{91.6} \pm 0.4$ | $\mathbf{85.2} \pm 0.1$ | $58.4 \pm 0.6$ |
| **CIFAR10** | Random | $15.7 \pm 0.9$ | $16.3 \pm 0.4$ | $13.1 \pm 0.6$ | $28.2 \pm 0.2$ |
| | SimCLR | $65.2 \pm 0.2$ | $67.8 \pm 0.2$ | $65.2 \pm 0.2$ | $49.8 \pm 2.8$ |
| | VicReg | $\mathbf{68.8} \pm 0.2$ | $\mathbf{69.6} \pm 0.2$ | $\mathbf{68.2} \pm 0.4$ | $\mathbf{54.3} \pm 0.7$ |
| | MoCo | $53.3 \pm 1.3$ | $56.4 \pm 1.6$ | $54.0 \pm 2.0$ | $35.0 \pm 2.8$ |
| | VAE | $24.7 \pm 0.4$ | $30.3 \pm 0.4$ | $25.6 \pm 0.5$ | $23.4 \pm 0.7$ |
| | $\beta$-VAE ($\beta = 1.2$) | $24.4 \pm 0.4$ | $29.8 \pm 0.2$ | $25.1 \pm 0.4$ | $23.8 \pm 0.4$ |
| | CR-VAE | $24.7 \pm 0.4$ | $30.4 \pm 0.1$ | $25.4 \pm 0.4$ | $23.9 \pm 0.8$ |
| | SimVAE | $\mathbf{36.4} \pm 0.0$ | $\mathbf{45.5} \pm 0.2$ | $\mathbf{42.8} \pm 0.0$ | $\mathbf{34.7} \pm 0.5$ |

Table 1: Top-1% self-supervised CA ($\uparrow$) for MNIST, FashionMNIST, CIFAR10, and Celeb-A (gender classification) using a linear probe (LP), MLP probe (MP), k-Nearest Neighbors (KNN), and Gaussian Mixture Model (GMM) classification methods; We report mean and standard errors over three runs; Bold indicate best scores in each method class: generative (blue), discriminative methods (red).

## 6 RESULTS

**Content classification** Table 1 reports the downstream classification and clustering accuracy across datasets. For Celeb-A, the only multi-attribute dataset, we report values for gender prediction. Table 1 shows that SimVAE is comparable to or outperforms *generative* baselines on supervised and unsupervised learning metrics on simple datasets. This is amplified when moving towards natural image data where we observe a significant improvement in performance over all VAE methods including the self-supervised approach, CR-VAE ($+4\%$ for Celeb-A, $+15\%$ for CIFAR10).

Table 1 also allows for the comparison of SimVAE with representations learned through popular *discriminative* methods. While a significant gap remains to be bridged between SimVAE and VicReg or SimCLR, the proposed method materially reduces the gap ($\Delta$) by *approximately half* for both Celeb-A ($\Delta = 7\% \rightarrow \Delta = 3.1\%$) and CIFAR10 ($\Delta = 39.2\% \rightarrow \Delta = 24.1\%$).

**Stylistic classification** We further analyse the learned representations by predicting multiple different attributes from the same representation. We leverage the broad range of Celeb-A facial attributes and evaluate attribute classification of SimVAE and baseline models across 20 tasks. Figure 3 shows that, on average, SimVAE outperforms *all generative and discriminative* baselines across these tasks, with a performance increase of more than $3\%$ and $15\%$ over generative and discriminative baselines, respectively, for individual tasks such as hair color prediction. This quantitatively supports our hypothesis that discriminative methods lose *stylistic* information (since embeddings that share the same *content* are collapsed together), which may be important for some downstream tasks.

|         | LP-CA            | MP-CA            |
|---------|------------------|------------------|
| Random  | $51.2 \pm 0.1$   | $52.9 \pm 0.4$   |
| SimCLR  | $\mathbf{66.8} \pm 0.2$ | $\mathbf{65.3} \pm 0.1$ |
| VicReg  | $64.2 \pm 0.5$   | $59.4 \pm 0.3$   |
| VAE     | $72.1 \pm 0.3$   | $78.2 \pm 0.5$   |
| $\beta$-VAE | $67.6 \pm 0.4$ | $72.5 \pm 0.7$   |
| CR-VAE  | $67.2 \pm 0.4$   | $75.9 \pm 0.4$   |
| SimVAE  | $\mathbf{75.1} \pm 0.3$ | $\mathbf{80.9} \pm 0.5$ |

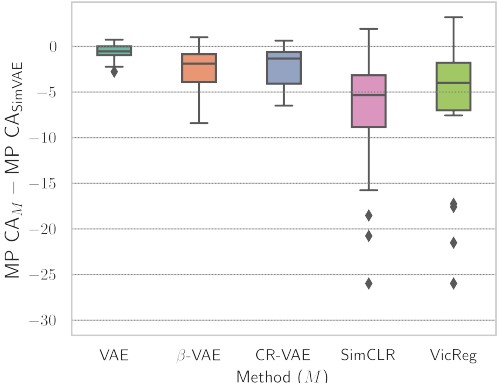

Figure 3: (*left*) Celeb-A hair color prediction (mean and std err for 3 runs). (*right*) Celeb-A multi-attribute prediction using MP: we show the average *performance gap* ($\Delta$) between SimVAE and baselines across 20 attribute prediction tasks and 3 random seeds. On average, SimVAE outperforms all other models ($\Delta < 0$), in particular discriminative SSL methods (SimCLR, VicReg).

A detailed analysis for each individual task is reported in appendix A.4. These findings support the observations made in Figure 1. We visualise image reconstructions for MNIST and FashionMNIST using VAEs' decoder and decoder networks trained post-hoc on frozen features learned through discriminative methods. This qualitatively suggests that more stylistic information (i.e., object position & size) is discarded by discriminative approaches that generative frameworks, such as SimVAE, are able to retain.

**Image Generation** While the focus of our work is not the generative quality relateive to prior work, rather to improve representational quality through a generative framework, we show randomly generated images using SimVAE as well as generative quality metrics in appendix A.4. We observe minor but significant FID score and reconstruction error improvements when using SimVAE with respect to other VAE-based methods for FashionMNIST, Celeb-A and CIFAR10 datasets.

## 7 DISCUSSION

We introduce the SimVAE training objective, based on the ELBO for a graphical model that embodies the assumptions implicit in a variety of discriminative self-supervised methods. Our results validate the assumptions in this latent variable model and demonstrate the efficacy of SimVAE relative to previous generative VAE-based approaches, including CR-VAE that aims for comparable latent structure. SimVAE demonstrably reduces the performance gap to discriminative self-supervised objectives, including those based on the InfoNCE objective.

SimVAE offers a more principled approach to modeling sets of semantically related observations, facilitating the simultaneous representation of both content and style information, and taking a positive step towards fully task-agnostic representations. Additionally, the posterior provides an estimate of uncertainty, which may be important for critical downstream tasks, and the prior allows for explicit design choices, offering the future prospect of separating latent factors to achieve disentangled representations.

While we consider SimVAE to be a positive advancement in representation learning, challenges remain in bridging the gap between generative and discriminative methods. Previous research shows that leveraging more complex model architectures, e.g. NVAE (Vahdat & Kautz, 2020), StyleGAN (Karras et al., 2019), and CR-VAE (Sinha & Dieng, 2021), can significantly enhance the ability of generative models. In this work, we hold the model architecture constant for fair comparison of the loss functions, but the additional complexity of generative methods and the increased information that representations are required to retain, may require more expressive architectures (e.g. Dosovitskiy et al. (2020)). Further, we note an increased number of augmentations tend to improve discriminative methods but not necessarily generative approaches (e.g. see appendix A.4), suggesting a direction for future investigation.

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
