# A APPENDIX

## A.1 RELATIONSHIP BETWEEN INFONCE REPRESENTATIONS AND PMI

For data sampled $x \sim p(x)$ and augmentations $x' \sim p_\tau(x'|x)$ sampled according to a synthetic augmentation strategy, Oord et al. (2018) show that the InfoNCE objective for a particular sample $x$ is optimised if their respective representations $z$, $z'$ satisfy

$$\exp\{sim(z, z')\} = c\, \frac{p(x,x')}{p(x)p(x')}, \tag{8}$$

where $sim(\cdot, \cdot)$ is the similarity function (e.g. dot product), and $c$ is a proportionality constant. We note that $c$ may differ arbitrarily for each $x$ and might, more generally, be considered an arbitrary function of $x$, but for simplicity we consider the case for a particular $x$. Note also that $c$ is strictly positive since it is the ratio of a (positive) exponential term and a (non-negative) probability ratio. Accordingly, representations satisfy

$$sim(z, z') = \text{PMI}(x, x') + c', \tag{9}$$

where $c' = \log c \in \mathbb{R}$ and $\text{PMI}(x, x')$ is the *pointwise mutual information* between data and augmentations. Pointwise mutual information (PMI) is a term from information theory that reflects the probability of events occurring jointly versus independently, which, for an arbitrary sample and arbitrary augmentation is given by:

$$\text{PMI}(x, x') \doteq \log \frac{p(x, x')}{p(x)p(x')} = \log \frac{p_\tau(x'|x)}{p(x')}. \tag{10}$$

Since $p_\tau(x'|x) = 0$ whenever $x'$ is *not* an augmentation of $x$, if augmentations of different data samples do not coincide (e.g. two original images cannot be augmented in different ways to give the same augmented image, as typically assumed), the marginal $p(x') = \int_x p_\tau(x'|x)p(x)$ is given by a single term $p_\tau(x'|x^*)p(x^*)$, where $x^*$ is the original sample from which $x'$ was augmented. Thus

$$\frac{p_\tau(x'|x)}{p(x')} = \frac{p_\tau(x'|x)}{p_\tau(x'|x^*)p(x^*)} = \begin{cases} 1/p(x^*) & \text{if } x^* = x \text{ (i.e. } x' \text{ is an augmentation of } x) \\ 0 & \text{otherwise;} \end{cases} \tag{11}$$

and $\text{PMI}(x, x') = -\log p(x) \geq 0$ or $\text{PMI}(x, x') = -\infty$, respectively.

If the main objective were to accurately approximate PMI (subject to a constant $c'$) in Eq. 9, e.g. to approximate *mutual information*, or if representation learning *depended* on it, then, at the very least, the domain of $sim(\cdot, \cdot)$ must span its range of values. For a typical dataset, PMI values range from $-\infty$ for negative samples to small positive values (e.g. 0-20) for positive samples. Despite this, the popular *bounded* cosine similarity function ($cossim(z, z') = \frac{z^T z}{||z||_2||z'||_2} \in [-1, 1]$) is found to significantly outperform the *unbounded* dot product even though the cosine similarity function necessarily cannot span the range required to reflect true PMI values, which the dot product can. This strongly suggests that representation learning does not require representations to capture PMI values, or for the overall loss function to approximate mutual information. Instead the cosine similarity-*restricted* InfoNCE objective is optimised if representations of a data sample and its augmentations are fully aligned ($cossim(z, z') = 1$) and representations of dissimilar data being maximally spread $cossim(z, z') = -1$, since these minimise the difference to the true PMI values for positive and negative samples (described above). Constraints such as the dimensionality of the representation space vs the number of samples may prevent these revised theoretical optima being fully achieved but show that the loss function is optimised by clustering representations of augmentations and the sample they are derived from and separating representations otherwise.

We note that our theoretical justification for representations *not* capturing PMI is supported by the empirical observation that closer approximations of mutual information do not appear to improve representations (Tschannen et al., 2020). Also, more recent contrastive self-supervised methods increase the cosine similarity between semantically related data but spread apart representation the without negative sampling of InfoNCE, yet outperform the InfoNCE objective despite having no obvious relationship to PMI (Grill et al., 2020; Bardes et al., 2022).

## A.2 Objective Derivation

Let $\mathbf{x} = \{x^1, ..., x^J\}$ be a set of $J$ semantically related samples and $\theta = \{\theta_x, \theta_z, \pi\}$, $\phi = \{\phi_z, \phi_y\}$ be parameters of the generative model, $p_\theta(\mathbf{x}, \mathbf{z})$, and approximate posterior, $q_\phi(\mathbf{z}|\mathbf{x})$, respectively. We derive the Evidence Lower Bound (ELBO) used as the SimVAE optimization objective (§4.2) as:

$$
\begin{aligned}
\min_\theta D_{\mathrm{KL}}[\, p(\mathbf{x}) \,\|\, p_\theta(\mathbf{x}) \,] &= \max_\theta \mathbb{E}_{\mathbf{x}}\big[\log p_\theta(\mathbf{x})\big] \\
&= \max_{\theta,\phi} \mathbb{E}_{\mathbf{x}}\Big[\int_{\mathbf{z}} q_{\phi_z}(\mathbf{z}|\mathbf{x}) \log p_\theta(\mathbf{x})\Big] \\
&= \max_{\theta,\phi} \mathbb{E}_{\mathbf{x}}\Big[\int_{\mathbf{z}} q_{\phi_z}(\mathbf{z}|\mathbf{x}) \log \tfrac{p_{\theta_x}(\mathbf{x}|\mathbf{z})p_{\theta_z}(\mathbf{z})}{p_\theta(\mathbf{z}|\mathbf{x})} \tfrac{q_\phi(\mathbf{z}|\mathbf{x})}{q_{\phi_z}(\mathbf{z}|\mathbf{x})}\Big] \\
&= \max_{\theta,\phi} \mathbb{E}_{\mathbf{x}}\Big[\int_{\mathbf{z}} q_{\phi_z}(\mathbf{z}|\mathbf{x}) \log \tfrac{p_{\theta_x}(\mathbf{x}|\mathbf{z})p_{\theta_z}(\mathbf{z})}{q_{\phi_z}(\mathbf{z}|\mathbf{x})}\Big] + D_{\mathrm{KL}}[\, q_\phi(\mathbf{z}|\mathbf{x}) \,\|\, p_\theta(\mathbf{z}|\mathbf{x}) \,] \\
&\geq \max_{\theta,\phi} \mathbb{E}_{\mathbf{x}}\Big[\int_{\mathbf{z}} q_{\phi_z}(\mathbf{z}|\mathbf{x})\Big\{\log \tfrac{p_{\theta_x}(\mathbf{x}|\mathbf{z})}{q_{\phi_z}(\mathbf{z}|\mathbf{x})} + \log p_{\theta_z}(\mathbf{z})\Big\}\Big] && (cf\ \text{Eq. 2}) \\
&= \max_{\theta,\phi} \mathbb{E}_{\mathbf{x}}\Big[\int_{\mathbf{z}} q_{\phi_z}(\mathbf{z}|\mathbf{x})\Big\{\log \tfrac{p_{\theta_x}(\mathbf{x}|\mathbf{z})}{q_{\phi_z}(\mathbf{z}|\mathbf{x})} + \log \int_y p_{\theta_z}(\mathbf{z}|y)p_\pi(y)\Big\}\Big] && (*) \\
&= \max_{\theta,\phi} \mathbb{E}_{\mathbf{x}}\Big[\int_{\mathbf{z}} q_{\phi_z}(\mathbf{z}|\mathbf{x})\Big\{\log \tfrac{p_{\theta_x}(\mathbf{x}|\mathbf{z})}{q_{\phi_z}(\mathbf{z}|\mathbf{x})} + \log \int_y p_{\theta_z}(\mathbf{z}|y)p_\pi(y)\tfrac{q_{\phi_y}(y|\mathbf{z})}{q_{\phi_y}(y|\mathbf{z})}\Big\}\Big] \\
&\geq \max_{\theta,\phi} \mathbb{E}_{\mathbf{x}}\Big[\int_{\mathbf{z}} q_{\phi_z}(\mathbf{z}|\mathbf{x})\Big\{\log \tfrac{p_{\theta_x}(\mathbf{x}|\mathbf{z})}{q_{\phi_z}(\mathbf{z}|\mathbf{x})} + \int_y q_{\phi_y}(y|\mathbf{z}) \log \tfrac{p_{\theta_z}(\mathbf{z}|y)p_\pi(y)}{q_{\phi_y}(y|\mathbf{z})}\Big\}\Big]
\end{aligned}
$$

The line indicated (*) is the ELBO if $y$ is continuous and $p(z|y)$ and $p(y)$ are chosen so that the final integral is tractable, e.g. all Gaussian. Where the integral is intractable, we continue with a further lower bound and introduce the approximate posterior $q(y|z)$. For explanatory purposes, we consider the case $J = 2 : \mathbf{x} = \{x, x'\}$. With a mean-field assumption $q(z, z'|x, x') = q(z|x)q(z'|x')$, we reach the following ELBO formulation:

$$
\begin{aligned}
\min_\theta & D_{\mathrm{KL}}[\, p(\mathbf{x}) \,\|\, p_\theta(\mathbf{x}) \,] \\
&\geq \max_{\theta,\phi_z} \mathbb{E}_{\mathbf{x}}\Big[\int_{z,z'} q_{\phi_z}(z, z'|x, x')\Big\{\log \tfrac{p_{\theta_x}(x,x'|z,z')}{q_{\phi_z}(z,z'|x,x')} + \log \int_y p_{\theta_z}(z, z'|y)p_\pi(y)\Big\}\Big] && (*) \\
&\geq \max_{\theta,\phi_z} \mathbb{E}_{\mathbf{x}}\Big[\underbrace{\int_z q_{\phi_z}(z|x)\log p_{\theta_x}(x|z)}_{[1]} - \underbrace{\int_z q_{\phi_z}(z|x)\log q_{\phi_z}(z|x)}_{[2]} \\
&\qquad\qquad + \underbrace{\int_{z'} q_{\phi_z}(z'|x')\log p_{\theta_x}(x'|z')}_{[3]} - \underbrace{\int_{z'} q_{\phi_z}(z'|x')\log q_{\phi_z}(z'|x')}_{[4]} \\
&\qquad\qquad + \underbrace{\int_{z,z'} q_{\phi_z}(z|x)q_{\phi_z}(z'|x')\int_y q_{\phi_y}(y|z, z') \log p_{\theta_z}(z|y)p_{\theta_z}(z'|y)\tfrac{p_\pi(y)}{q_{\phi_y}(y|z,z')}}_{[5]}\Big] && (12)
\end{aligned}
$$

Terms of the SimVAE objective in Equation (12) are analogous to those of the standard ELBO: [1] & [3] are commonly referred to as (negative) *reconstruction error*, [2] & [4] are *entropy* of the approximate posterior $H(q_\phi(\mathbf{z}|\mathbf{x}))$. Terms [1-4] are equivalent to terms found in the standard ELBO for each of the ($J = 2$) related samples. Term [5] derives naturally from the hierarchy of SimVAE and defines cluster structure of $p(z, z')$, between representations of semantically related data, in terms of $p(z|y)$ and $p(y)$. Algorithm 1 provides an overview of the computational steps required for the training of the SimVAE evidence lower bound and details the steps required for the computation of [1]/[3], [2]/[4] & [5] referred to as the *rec*, *H*, *prior* terms respectively. As our experimental setting considers augmentations as semantically related samples, algorithm 1 incorporates a preliminary step to augment data samples.

## A.3 EXPERIMENTAL DETAILS

### A.3.1 DATASETS

**MNIST** The MNIST dataset (LeCun, 1998) gathers 60'000 training and 10'000 testing images representing digits from 0 to 9 in various caligraphic styles. Images were kept to their original 28x28 pixel resolution and were binarized. The 10-class digit classification task was used for evaluation.

**FashionMNIST** The FashionMNIST dataset (Xiao et al., 2017) is a collection of 60'000 training and 10'000 test images depicting Zalando clothing items (i.e., t-shirts, trousers, pullovers, dresses, coats, sandals, shirts, sneakers, bags and ankle boots). Images were kept to their original 28x28 pixel resolution. The 10-class clothing type classification task was used for evaluation.

**CIFAR10** The CIFAR10 dataset (Krizhevsky et al., 2009) offers a compact dataset of 60,000 (50,000 training and 10,000 testing images) small, colorful images distributed across ten categories including objects like airplanes, cats, and ships, with various lighting conditions. Images were kept to their original 32x32 pixel resolution.

**Celeb-A** The Celeb-A dataset (Liu et al., 2015) comprises a vast collection of celebrity facial images. It encompasses a diverse set of 183'000 high-resolution images (i.e., 163'000 training and 20'000 test images), each depicting a distinct individual. The dataset showcases a wide range of facial attributes and poses and provides binary labels for 40 facial attributes including hair & skin color, presence or absence of attributes such as eyeglasses and facial hair. Each image was cropped and resized to a 64x64 pixel resolution. Attributes referring to hair color were aggregated into a 5-class attribute (i.e., bald, brown hair, blond hair, gray hair, black hair). Images with missing or ambiguous hair color information were discarded at evaluation.

All datasets were sourced from Pytorch's dataset collection.

### A.3.2 DATA AUGMENTATION STRATEGY

Taking inspiration from SimCLR's (Chen et al., 2020) augmentation strategy which highlights the importance of random image cropping and color jitter on downstream performance, our augmentation strategy includes random image cropping, random image flipping and random color jitter. The color augmentations are only applied to the non gray-scale datasets (i.e., CIFAR10 (Krizhevsky et al., 2009) & Celeb-A dataset (Liu et al., 2015)). Due to the varying complexity of the datasets we explored, hyperparameters such as the cropping strength were adapted to each dataset to ensure that semantically meaningful features remained after augmentation. The augmentation strategy hyperparameters used for each dataset are detailed in table 2.

| Dataset | Crop | | Vertical Flip | Color Jitter | | |
|---------|------|------|---------------|------|-----|-------|
| | scale | ratio | prob. | b-s-c | hue | prob. |
| MNIST | 0.4 | [0.75,1.3] | 0.5 | - | - | - |
| Fashion | 0.4 | [0.75,1.3] | 0.5 | - | - | - |
| CIFAR10 | 0.6 | [0.75,1.3] | 0.5 | 0.8 | 0.2 | 0.8 |
| Celeb-A | 0.6 | [0.75,1.3] | 0.5 | 0.8 | 0.2 | 0.8 |

Table 2: Data augmentation strategy for each dataset: (from left to right) cropping scale, cropping ratio, probability of vertical and horizontal flipping, brightness-saturation-contrast jitter strength, hue jitter strength, probability of color jitter

### A.3.3 TRAINING IMPLEMENTATION DETAILS

This section contains all details regarding the architectural and optimization design choices used to train SimVAE and all baselines. Method-specific hyperparameters are also reported below.

**Network Architectures** The encoder network architectures used for SimCLR, MoCo, VicReg, and VAE-based approaches including SimVAE for simple (i.e., MNIST, FashionMNIST ) and complex

datasets (i.e., CIFAR10, Celeb-A) are detailed in table 3a, table 4a respectively. Generative models which include all VAE-based methods also require decoder networks for which the architectures are detailed in table 3b and table 4b. The latent dimensionality for MNIST and FashionMNIST is fixed at 10 and increased to 64 for the Celeb-A and CIFAR10 datasets. The encoder and decoder architecture networks are kept constant across methods including the latent dimensionality to ensure a fair comparison.

| Layer Name | Output Size | Block Parameters |
|---|---|---|
| fc1 | 500 | 784x500 fc, relu |
| fc2 | 500 | 500x500 fc, relu |
| fc3 | 2000 | 500x2000 fc, relu |
| fc4 | 10 | 2000x10 fc |

(a) Encoder

| Layer Name | Output Size | Block Parameters |
|---|---|---|
| fc1 | 2000 | 10x2000 fc, relu |
| fc2 | 500 | 2000x500 fc, relu |
| fc3 | 500 | 500x500 fc, relu |
| fc4 | 784 | 500x784 fc |

(b) Decoder

Table 3: Multi-layer perceptron network architectures used for MNIST & FashionMNIST training

| Layer Name | Output | Block Parameters |
|---|---|---|
| conv1 | 32x32 | 4x4, 16, stride 1 |
| | | batchnorm, relu |
| | | 3x3 maxpool, stride 2 |
| conv2_x | 32x32 | 3x3, 32, stride 1 |
| | | 3x3, 32, stride 1 |
| conv3_x | 16x16 | 3x3, 64, stride 2 |
| | | 3x3, 64, stride 1 |
| conv4_x | 8x8 | 3x3, 128, stride 2 |
| | | 3x3, 128, stride 1 |
| conv5_x | 4x4 | 3x3, 256, stride 2 |
| | | 3x3, 256, stride 1 |
| fc | 64 | 4096x64 fc |

(a) Encoder

| Layer Name | Output | Block Parameters |
|---|---|---|
| fc | 256x4x4 | 64x4096 fc |
| conv1_x | 8x8 | 3x3, 128, stride 2 |
| | | 3x3, 128, stride 1 |
| conv2_x | 16x16 | 3x3, 64, stride 2 |
| | | 3x3, 64, stride 1 |
| conv3_x | 32x32 | 3x3, 32, stride 2 |
| | | 3x3, 32, stride 1 |
| conv4_x | 64x64 | 3x3, 16, stride 2 |
| | | 3x3, 16, stride 1 |
| conv5 | 64x64 | 5x5, 3, stride 1 |

(b) Decoder

Table 4: Resnet18 network architectures used for CIFAR10 & Celeb-A datasets

**Optimisation & Hyper-parameter tuning** All methods were trained using an Adam optimizer until training loss convergence. A learning rate tuning was performed for each method independently across the range $1e^{-3}$ to $8e^{-5}$. A fixed batch size of 128 was used across methods and datasets. The $\beta, \tau, \lambda$ parameters for the $\beta$-VAE, SimCLR and CRVAE methods were tuned across the [0.1,0.2,0.5], [0.1,0.5,1.0] and [0.01,0.1,1.0] ranges respectively based on downstream performance. $\beta = 0.1$, $\lambda = 0.01$ were selected and $\tau = 1.0, \tau = 0.5$ were chosen for simple and natural datasets respectively. The likelihood probability variance for VAE-based methods including SimVAE was kept to $\sigma^2 = 1.0$ and the prior probability, $p(z|y)$, variance parameter for SimVAE was tuned based on downstream performance and fixed to 0.005, 0.003, 0.005, 0.005 for MNIST, FashionMNIST, CIFAR10, and Celeb-A respectively.

### A.3.4 EVALUATION IMPLEMENTATION DETAILS

Following common practices (Chen et al., 2020), downstream performance is assessed using a linear probe, a multi-layer perceptron probe, a k-nearest neighbors (kNN) algorithm, and a Gaussian mixture model (GMM). The linear probe consists of a fully connected layer whilst the mlp probe consists of two fully connected layers with a relu activation for the intermediate layer. Both probes were trained using an Adam optimizer with a learning rate of 3e-4 for 200 epochs with batch size fixed to 128. Scikit-learn's Gaussian Mixture model with a full covariance matrix and 200 initialization was

fitted to the representations using the ground truth cluster number. The kNN algorithm from Python's Scikit-learn library was used with k spanning from 1 to 15 neighbors. The best performance was chosen as the final performance measurement. No augmentation strategy was used at evaluation.

### A.3.5 GENERATION PROTOCOL

In this section, we detail the image generation protocol as well as the evaluation of the quality of the generated samples.

**Ad-hoc decoder training** VAE-based approaches, including SimVAE, are fundamentally generative methods aimed at approximating the logarithm of the marginal likelihood distribution, denoted as $\log p(x)$. In contrast, most traditional self-supervised methods adopt a discriminative framework without a primary focus on accurately modeling $p(x)$. However, for the purpose of comparing representations, and assessing the spectrum of features present in $z$, we intend to train a decoder model for SimCLR & VicReg models. This decoder model is designed to reconstruct images from the fixed representations initially trained with these approaches. To achieve this goal, we train decoder networks using the parameter configurations specified in Tables 3b and 4b, utilizing the mean squared reconstruction error as the loss function. The encoder parameters remain constant, while we update the decoder parameters using an Adam optimizer with a learning rate of $1e^{-4}$ until convergence is achieved (i.e. $\sim 200$ epochs).

**Conditional Image Generation** To allow for a fair comparison, all images across all methods are generated by sampling $z$ from a multivariate Gaussian distribution fitted to the training samples' representations. More precisely, each Gaussian distribution is fitted to $z$ conditioned on a label $y$. Scikit-Learn Python library Gaussian Mixture model function (with full covariance matrix) is used.

### A.4 ADDITIONAL RESULTS

#### A.4.1 SINGLE-TASK CLASSIFICATION

**Clustering metrics** Table 5 and table 6 report the normalized mutual information (NMI) and adjusted rank index (ARI) for the fitting of a GMM to latent representations $z$.

| Dataset | | Random | VAE | $\beta$-VAE | CR-VAE | SimVAE |
|---|---|---|---|---|---|---|
| **MNIST** | ARI | $21.5 \pm 1.4$ | $\mathbf{98} \pm \mathbf{0.1}$ | $93.7 \pm 0.9$ | $97.6 \pm 0.0$ | $94.2 \pm 0.0$ |
| | NMI | $46.1 \pm 1.3$ | $96.3 \pm 0.4$ | $96.6 \pm 0.4$ | $88.2 \pm 1.7$ | $\mathbf{97.1} \pm \mathbf{0.0}$ |
| **Fashion** | ARI | $28.7 \pm 0.6$ | $44.2 \pm 1.1$ | $44.7 \pm 0.2$ | $23.3 \pm 0.8$ | $\mathbf{55.7} \pm \mathbf{0.0}$ |
| | NMI | $51.5 \pm 0.2$ | $66.7 \pm 0.7$ | $66.4 \pm 0.4$ | $46.1 \pm 2.2$ | $\mathbf{76.8} \pm \mathbf{0.2}$ |
| **Celeb-A** | ARI | $3.4 \pm 0.3$ | $5.7 \pm 0.2$ | $6.2 \pm 0.7$ | $6.6 \pm 0.9$ | $2.6 \pm 0.7$ |
| | NMI | $4.2 \pm 0.4$ | $3.9 \pm 0.2$ | $4.7 \pm 0.9$ | $5.0 \pm 0.7$ | $2.9 \pm 0.7$ |
| **CIFAR10** | ARI | $0.09 \pm 0.0$ | $0.7 \pm 0.2$ | $0.7 \pm 0.2$ | $0.9 \pm 0.1$ | $\mathbf{8.6} \pm \mathbf{0.3}$ |
| | NMI | $27.9 \pm 0.1$ | $17.7 \pm 0.5$ | $18.7 \pm 0.3$ | $18.9 \pm 0.1$ | $\mathbf{37.2} \pm \mathbf{0.4}$ |

Table 5: Normalized mutual information (NMI) and Adjusted Rank Index (ARI) for all generative methods and datasets; Average scores and standard errors are computed across three random seeds

#### A.4.2 MULTI-TASK CLASSIFICATION

Figure 4 reports the average classification accuracy using a MLP probe across 3 random seeds for the prediction of each of 20 Celeb-A facial attributes for SimVAE, generative and discriminative baselines.

**Augmentation protocol strength ablation** Figure 5 reports the downstream CA across methods for various augmentations stategy. More precisely, we progressively increase the cropping scale and color jitter amplitude. Unsurprisingly (Chen et al., 2020), discriminative methods exhibit high sensitivity to the augmentation strategy with stronger disruption leading to improved content prediction. The

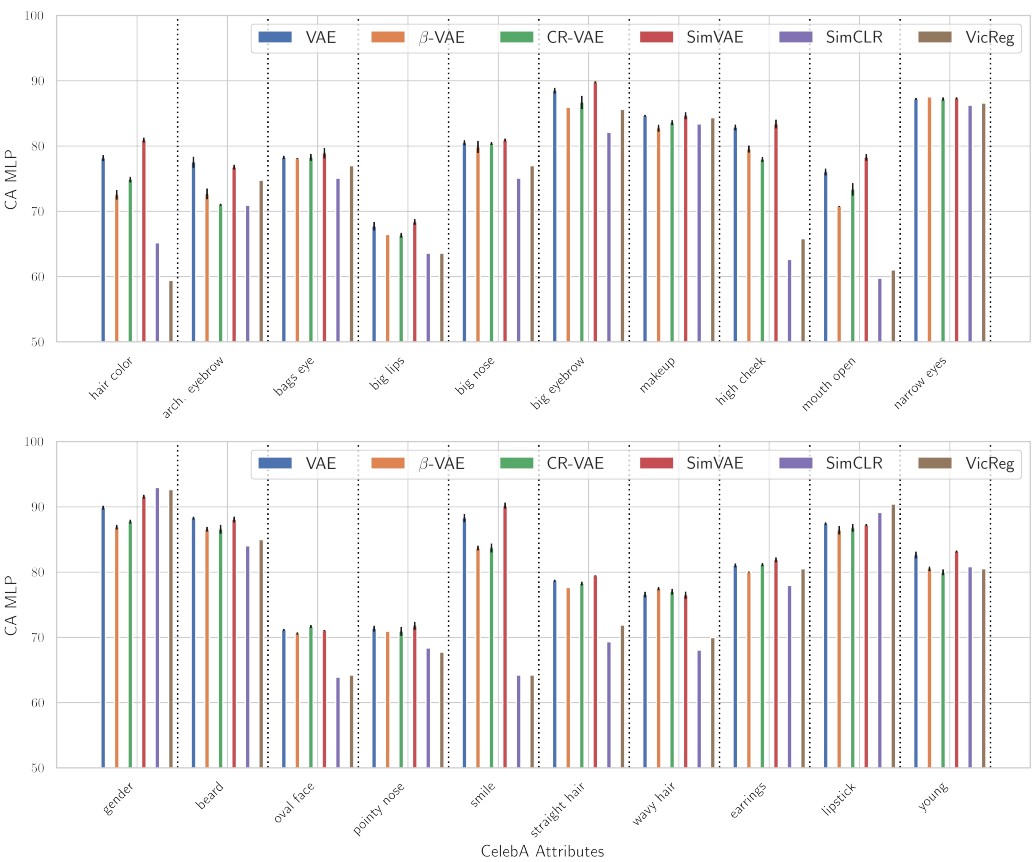

Figure 4: Celeb-A 20 facial attributes prediction using a MP. Average scores and standard errors are reported across 3 random seeds.

| Dataset | | MoCo | VicReg | SimCLR |
|---|---|---|---|---|
| **MNIST** | ARI | $58.3 \pm 3.8$ | $73.1 \pm 1.8$ | $\mathbf{95.3} \pm \mathbf{1.2}$ |
| | NMI | $71.4 \pm 2.5$ | $86.2 \pm 1.2$ | $\mathbf{97.5} \pm \mathbf{0.6}$ |
| **Fashion** | ARI | $30.9 \pm 0.5$ | $37.1 \pm 1.3$ | $\mathbf{50.3} \pm \mathbf{1.9}$ |
| | NMI | $50.4 \pm 0.6$ | $64.5 \pm 0.7$ | $\mathbf{71.2} \pm \mathbf{1.0}$ |
| **Celeb-A** | ARI | $-$ | $\mathbf{18.7} \pm \mathbf{0.8}$ | $0.0 \pm 0.1$ |
| | NMI | $-$ | $\mathbf{24.3} \pm \mathbf{0.3}$ | $0.0 \pm 0.0$ |
| **CIFAR10** | ARI | $27.2 \pm 1.0$ | $31.2 \pm 0.2$ | $\mathbf{49.6} \pm \mathbf{1.3}$ |
| | NMI | $16.5 \pm 0.4$ | $\mathbf{53.4} \pm \mathbf{0.1}$ | $26.9 \pm 0.8$ |

Table 6: Normalized mutual information (NMI) and Adjusted Rank Index (ARI) for all discriminative baselines and datasets; Average scores and standard errors are computed across three random seeds

opposite trend is observed with vanilla generative methods where reduced variability amongst the data leads to increased downstream performance. Interestingly, SimVAE is robust to augmentation protocol and performs comparably across settings.

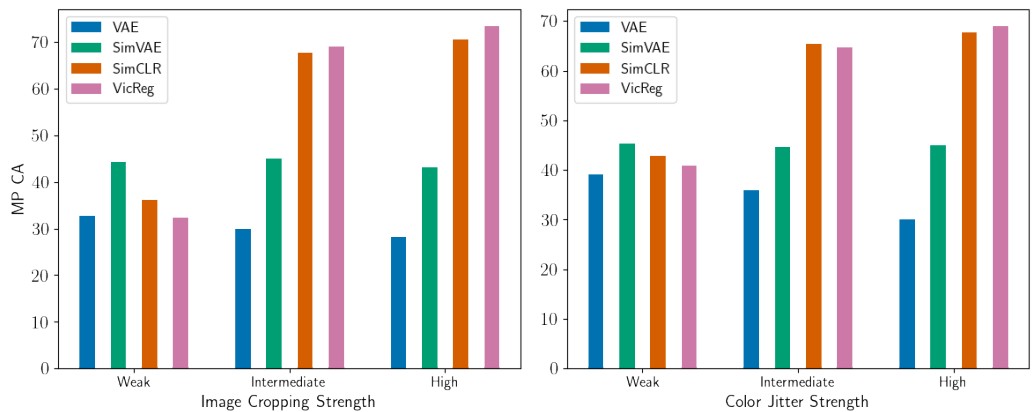

Figure 5: Ablation experiment across the number of augmentations considered during training of the SimVAE model using the MNIST (left) and FashionMNIST (right) datasets. Two, four, six and eight augmentations were considered. The average and standard deviation of the downstream classification accuracy using KNN and GMM probes are reported across three seeds.

**# of augmentation ablation** Figure 6 reports the downstream classification accuracy for increasing numbers of augmentations considered simultaneously during the training of SimVAE for MNIST and FashionMNIST datasets. A larger number of augmentations result in a performance increase up to a certain limit (i.e., 6-8 augmentations). Further exploration is needed to understand how larger sets of augmentations can be effectively leveraged potentially by allowing for batch size increase. Due to computational limitations, CIFAR10 & Celeb-A experiments rely on pairs of augmentations only.

**Likelihood $p(x|z)$ variance ablation** We explore the impact of the likelihood, $p(x|z)$, variance, $\sigma^2$, across each pixel dimension on the downstream performance using the CIFAR10 dataset. Appendix A.4.2 highlights how the predictive performance is inversely correlated with the $\sigma^2$, highlighting how SimVAE's performance can further benefit from a reduction of $\sigma^2$.

### A.4.3 IMAGE GENERATION

In this section, we explore and report the quality of images generated through SimVAE and all considered baselines through visualisations (for VAE-based approaches only) and quantitative

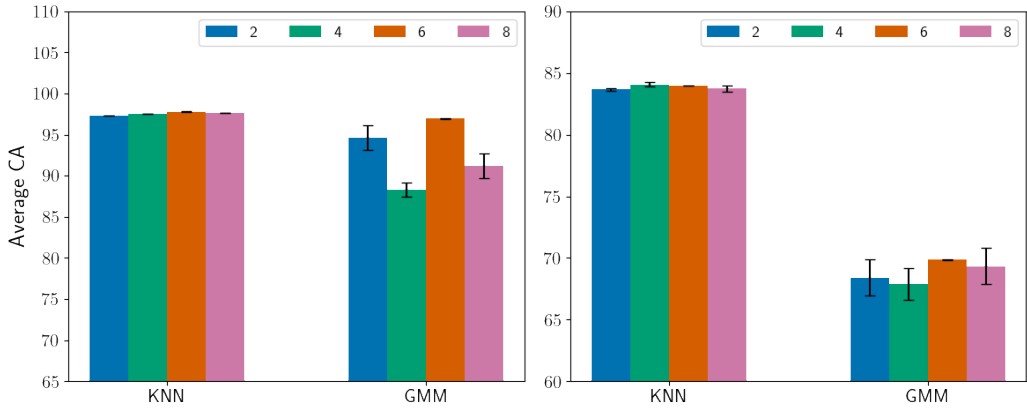

Figure 6: Ablation experiment across the number of augmentations considered during training of the SimVAE model using the MNIST (left) and FashionMNIST (right) datasets. Two, four, six and eight augmentations were considered. The average and standard deviation of the downstream classification accuracy using KNN and GMM probes are reported across three seeds. Batch size of 128 for all reported methods and number of augmentations.

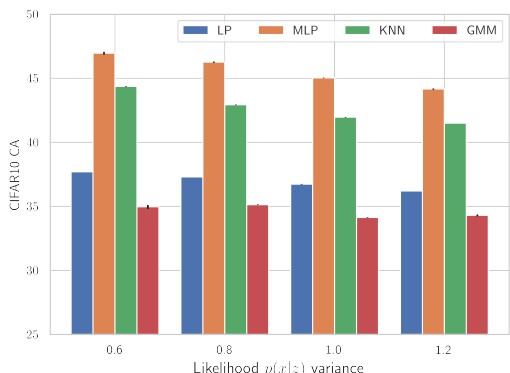

Figure 7: Ablation experiment across the likelihood, $p(x|z)$, variance. The average and standard errors of CIFAR10 classification accuracy (CA) using a linear probe (LP), mlp probe (MP), k-NN algorithm (KNN) and Gaussian mixture model (GMM) are reported across three random seeds.

measurements.

**Generated Images** Figures 8 and 9 report examples of randomly generated images for each digit class and clothing item using the SimVAE trained on MNIST and FashionMNIST respectively.

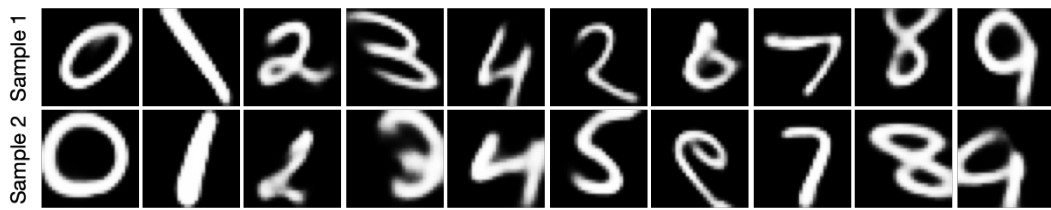

Figure 8: Conditional sampling for each one of the MNIST digit using pre-trained SimVAE model

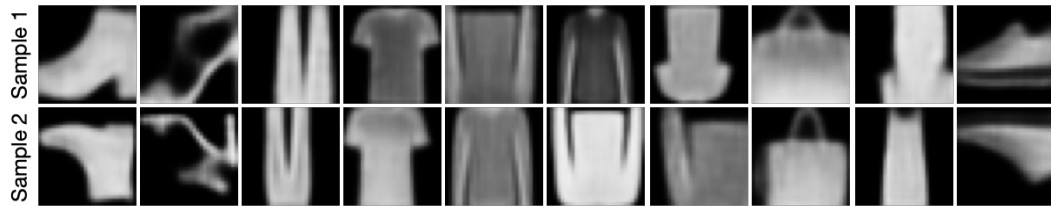

Figure 9: Conditional sampling for each one of the FashionMNIST clothing type using pre-trained SimVAE model

|  |  | RE | FID | NLL |
|---|---|---|---|---|
| MNIST | VAE | $11.3 \pm 0.1$ | $\mathbf{150.1} \pm 0.2$ | $5703.8 \pm 0.1$ |
|  | $\beta$-VAE | $11.3 \pm 0.0$ | $155.3 \pm 0.5$ | $5703.9 \pm 0.0$ |
|  | CR-VAE | $11.8 \pm 0.0$ | $153.0 \pm 0.9$ | $5705.2 \pm 0.1$ |
|  | SimVAE | $\mathbf{11.2} \pm 0.0$ | $152.7 \pm 0.3$ | $5703.5 \pm 0.0$ |
| Fashion | VAE | $4.4 \pm 0.1$ | $99.4 \pm 0.6$ | $5696.5 \pm 0.1$ |
|  | $\beta$-VAE | $4.6 \pm 0.1$ | $99.9 \pm 0.7$ | $5696.7 \pm 0.1$ |
|  | CR-VAE | $4.3 \pm 0.0$ | $98.7 \pm 0.0$ | $5696.7 \pm 0.0$ |
|  | SimVAE | $\mathbf{3.4} \pm 0.1$ | $\mathbf{96.1} \pm 1.0$ | $5695.6 \pm 0.0$ |
| Celeb-A | VAE | $56.6 \pm 0.2$ | $162.9 \pm 2.8$ | — |
|  | $\beta$-VAE | $60.3 \pm 1.0$ | $163.8 \pm 2.3$ | — |
|  | CR-VAE | $57.4 \pm 0.1$ | $\mathbf{159.3} \pm 5.4$ | — |
|  | SimVAE | $\mathbf{35.3} \pm 0.2$ | $157.8 \pm 2.3$ | — |
| CIFAR10 | VAE | $\mathbf{21.4} \pm 0.2$ | $365.4 \pm 3.3$ | $22330.8 \pm 0.2$ |
|  | $\beta$-VAE | $22.3 \pm 0.2$ | $376.7 \pm 1.7$ | $22327.7 \pm 0.2$ |
|  | CR-VAE | $22.5 \pm 0.0$ | $374.4 \pm 0.4$ | $22327.3 \pm 0.8$ |
|  | SimVAE | $22.1 \pm 0.1$ | $\mathbf{349.9} \pm 2.1$ | $22327.3 \pm 0.2$ |

Table 7: Generation quality evaluation of all generative methods across three random seeds: (from left to right) mean squared reconstruction error (RE, ↓), fréchet inception distance (FID, ↓), negative log-likelihood (NLL, ↓)

**Generative quality** Table 7 reports the FID scores, reconstruction error and approximate negative log-likelihoods using 1000 importance-weighted samples for all generative baselines and SimVAE.