# OpenReview forum: "SimVAE: Narrowing the gap between Discriminative & Generative Self-Supervised Representation Learning"
_ICLR.cc/2024/Conference — Submitted to ICLR 2024_

### Official Review · Reviewer_xsXw · 2023-10-18

**Soundness:** 2 fair
**Presentation:** 2 fair
**Contribution:** 2 fair
**Rating:** 5
**Confidence:** 5

**Summary:**

The paper provides a Bayesian perspective on self-supervised learning making an explicit connection to VAE-based models, thus enabling to incorporate properties inherent to both discriminative and generative approaches. The proposed framework highlights an underlying probabilistic graphical model for self-supervised learning (SSL) and a corresponding ELBO-like objective. Experiments are conducted on MNIST, FashionMNIST, CIFAR10 and Celeb-A, against SSL and VAE-based baseline models. The results highlights (i) that the proposed approach is competitive on simple datasets in terms of discrimination performance with SSL, with the advantage of retaining both information about content and style thanks to the generative aspect and (ii) that there exists a gap in discriminative performance between the proposed approach (also VAE models) and SSL on natural images (CIFAR-10).

**Strengths:**

1. The problem of unifying SSL and generative approaches is relevant and timely (**Relevance**)
2. The paper is clear and well-written (**Clarity**)

**Weaknesses:**

1. The paper omits important related work [1-6]. At a minimum, a discussion about the similarities and differences should be included (**Quality**).
2. Parts of the paper, especially on the background of self-supervised learning, are overly simplified and imprecise (for instance regarding the classes of SSL approaches), please refer to [3] and [7] (**Quality**).
3. Some of the main claims of the paper are not well-supported, especially the ones about the unification between SSL and generative approaches. Please refer to the general analysis in [3] and [4]. The novelty and theoretical contribution is somewhat limited, perhaps lying in specializing the existing framework (GEDI in [3] and [4]) to the VAE setting (**Novelty**).
4. While the experimental analysis provides evidence on the benefits of the proposed unification, the conclusions drawn from the experiments are rather limited confirming what has been already observed partly in [8] and in [3,4] (**Significance/Novelty**). Perhaps, the authors should focus on deepening the analysis on the existing gap observed on natural images (CIFAR-10), in order to improve in terms of significance and novelty.
5. The experimental analysis is missing a comparison with other existing generative and discriminative models [3] and [4] (**Quality**).

**MINOR**

In Section 4.2, all conditional densities should be explicitly defined.

**References**

[1] Learning Symbolic Representations Through Joint Generative and Discriminative Training. ICLR Workshop NeSy-GeMs 2023

[2] Learning Symbolic Representations Through Joint Generative and Discriminative Training (Extended Abstract). IJCAI Workshop KBCG 2023

[3] GEDI: GEnerative and Discriminative Training for Self-Supervised Learning. arXiv 2022

[4] The Triad of Failure Modes and A Possible Way Out. arXiv 2023

[5] D2C: Diffusion-Decoding Models for Few-Shot Conditional Generation. NeurIPS 2021

[6] Guiding Energy-based Models via Contrastive Latent Variables. ICLR 2023

[7] A Cookbook of Self-Supervised Learning. arXiv 2023

[8] Self-Supervised Learning with Data Augmentations Provably Isolates Content From Style. NeurIPS 2021

**Questions:**

1. Can you please discuss the similarities and differences between the above-mentioned references, especially [3] and [4]?
2. What are the main reasons behind the existing gap between VAE-like and SSL models observed on CIFAR-10?
3. What is the equivalent of the notion of “content and style” in natural images?
4. Can you please provide the definition of the conditionals introduced in Section 4.2?
5. What is the advantage of having a decoder compared to [3], [4], as this introduces additional computation?

---

> ### Author Response · Authors · 2023-11-20
> **Rebuttal by Authors**
>
> * **"Omitting important related work"**: Thank you for raising these works.
>     * [1-4]: for clarity (and benefit of other reviewers), we note that these are highly related to one another by the same authors:
>         - [4] is a **workshop submission** which further investigates a graphical model proposed in [3]. It appeared **after the ICLR submission date** (due at a [NeurIPS 23 workshop](https://nips.cc/virtual/2023/80847#:~:text=We%20present%20a%20novel%20objective,to%20permutations%20of%20cluster%20assignments.)), so we omit this per ICLR guidelines.
>         - [1, 2] are a **workshop paper and abstract** based on [3], so we restrict comments to [3].
>         - [3] is on Arxiv (not yet cited) and not peer-reviewed, and we were unaware of it. (Please note [ICLR Guidelines](https://iclr.cc/Conferences/2024/ReviewerGuide): "Authors ... may be excused for not knowing about papers not published in peer-reviewed conference proceedings or journals, which includes papers ... on arXiv.") [3] is related to Sim-VAE, we provide a detailed comparison below and cite it in our work (section 2).
>     - [5] introduces a VAE with a *diffusion* prior and a discriminative SSL loss (see Eq 4, 5 & Fig 2). This does not compare closely to our generative model for SSL which _unifies_ existing discriminative SSL under a unique ELBO. [5] does combine SSL with VAEs, we therefore cite it as related work in section 2.
>     - [6] trains Energy Based Models with a contrastive component and is less comparable to our generative latent variable approach.
> * **"Similarity & differences between SimVAE and [3]"**: SimVAE and [3] both set out to improve self-supervised learning and develop a more principled approach by understanding existing existing SSL methods from a probabilistic perspective. We highlight main differences:
>     - Model:
>         - [3] proposes a **different** graphical model for each class of SSL method considered (Fig 1): contrastive (CL), cluster-based (CB) and negative-free (NF). We note that these are not *generative* models of the data $x$ (e.g. arrows point *away* from $x$) leading to objective functions distinct from SimVAE's.
>         - In SimVAE, we consider the distribution over representations ($z$) induced by several SSL methods and formalise the intuition in a **single** generative hierarchical latent variable model (LVM): $p_{\theta, \psi, \pi}(x)\!=\!\int_{y,z} p_\theta(x|z)p_\psi(z|y)p_\pi(y)$ (Fig 2). This model is general, and it is assumed that $y$ conditions *lower-variance* distributions $p(z|y)$, e.g. *clusters*, in the latent space. We claim that *all* of the SSL methods considered (and related methods, by extension) implicitly assume this *one* latent variable model (Fig 2), i.e. representations are learned by assuming that those of semantically related data share the same $y$ and are therefore clustered together, while clusters are kept apart/preventing *collapse*. Specifically:
>             - instance discrimination (ID) treats the data index $i$ as the latent $y$, hence representations of each sample and its augmentations form clusters. CL methods (e.g. SimCLR) are shown to approximate ID, so exploit the same latent assumption (not their relationship to *mutual information*, as often assumed). NF models, e.g. BYOL, impose similar instance-level latent clustering, and differ algorithmically in how representations are prevented from collapsing.
>             - CB methods (e.g. DeepCluster) directly perform clustering in the latent space, using a large (arbitrary) number of clusters indexed by $y$, relying on the *inductive bias* of the encoder to map representations of semantically related data close together, even at random initialisation (see DeepCluster).
>     - Objective:
>         - [3] proposes a new SSL objective, GEDI (Eq 10), that maximises the likelihood under an **energy-based model** (EBM), $p_\psi(x) \propto e^{u^\top enc(x)}$ and minimises several KL divergences derived from CB & NF methods. This *amalgamates* different aspects of SSL approaches and a model of $p(x)$. GEDI appears to assume (in $p_\psi(x)$ and $L_{DI}$ terms of Eq 10) that the number of classes $c$, for some ground truth "class" is known, which contrastive SSL methods do not assume.
>         - In SimVAE, the proposed **generative LVM** *unifies* the considered SSL methods under one probabilistic model and the proposed objective is the ELBO for that LVM derived from first principles.
>     - Implementation:
>         - [3] uses a 2-stage process to first train the EBM using SGLD *sampling* and then train a discriminative model.
>         - SimVAE follows a single optimisation process (e.g. SGD).
>         - [3] also introduces a novel augmentation strategy ("DAM") that improves results, as does the assumption that $c$ is known (Figs 3 and 7). These provide GEDI with additional samples and information relative to other SSL methods (inc SimVAE), obfuscating like-for-like comparison.

---

> > ### Author Response · Authors · 2023-11-20
> > **Rebuttal by Authors**
> >
> > * **"Oversimplification of the SSL landscape vs [3,7]"**:
> >     - Thank you for your feedback. As mentioned above, we updated the manuscript to cite [3] in section 1. [7] reviews SSL and proposes a partitioning of the field into 4 main families of algorithms: a) the deep metric familty (includes contrastive methods), b) the self-distillation family (includes BOYL, DINO), c) the canonical correlation analysis family (includes VicReg) and the mask image modelling family. Our contribution covers and unifies a), b), c) as detailed in section 4.1. We updated the manuscript to integrate mask image modelling to the SSL related work and explicitely mention that the integration of mask image modelling methods into the proposed framework is left for further work.
> > * **"Claims not well supported & novelty of theoretical contribution vs [3]"**:
> >     - Please see the above comparison between SimVAE and [3], showing that the SimVAE model and training objective differs materially to [3] and, we believe, offers novel theoretical insight.
> > * **"Conclusions from experiments limited vs [3,8] ... Missing comparison with [3]."**
> >     - [8] relates specifically to the separation of *style* and *content* in SSL and runs experiments on  synthetic data, whereas we propose a hierarchical generative latent variable model for SSL and run experiments on real data, so do not see a direct comparison between these experimental results. For example, Fig 1 in [8] does not share the same hierarchical structure as our model (Fig 2).
> >     - [3] please see the comparison to SimVAE above, which explains why experimental results are not fairly comparable. Also, SimVAE learns parameters of a latent variable model under variational inference (using the ELBO) and compares to similar VAE-based models to show the benefit of our assumptions, whereas [3] is a significantly different EBM-based method.
> > * **"Reasons for the performance gap between VAE-like and SSL models"**:
> >     - From our analysis, we conjecture that this may be due, as least in part, to the last pooling layer in standard ResNets. This layer sums the tensor representation over spatial dimensions.
> >         - For discriminative SSL this seems beneficial for downstream class prediction as representations of an object in different spatial locations are made similar, in line with the SSL objective.
> >         - For generative SSL, the non-invertability of this step loses information required for reconstruction, but removing this step causes representations of the same object to differ, harming the SSL objective of clustering them together.
> >     - Thus the pooling step in standard ResNets is useful for discriminative SSL but hampers generative SSL and we believe this suggests that a different choice of encoder architecture may be required to narrow the performance gap in future work.
> > * **"*content* and *style* in natural images"**:
> >     - we use these terms loosely to provide intuition and in keeping with works that consider the separation of content and style in images [e.g. 8 and $\beta$-VAE], where *content* typically refers to "object class" and *style* refers to attributes such as lighting or pose. (This distinction is subjective, depending on which latent attributes are important to a task.)
> > * **"definition of conditionals in §4.2"**:
> >     - §4.2 describes the general model, which can be used for different types of data for which different distributional choices might be made. For our experiments on image data, conditionals $p(x|z)$ and $p(z|y)$ are modelled by Gaussians. This was made more explicit in section 5.
> > * **"Advantage of decoder compared to [3]"**:
> >     - Our training objective is derived from the standard ELBO to learn the parameters of a generative model $p_{\theta, \psi, \pi}(x) = \int_{z}p_\theta(x|z)p_\psi(z|y)p_\pi(y)$. As in standard VAEs, $p_\theta(x|z)$ is parameterised with a neural network referred to as the "decoder" (by analogy to a deterministic auto-encoder). The distribution $p(x|z)$ and hence the "decoder" is therefore an essential component of the training objective as opposed to a design choice.
> >     - By comparison, the GEDI loss function (Eq 10) in [3] is not derived from the ELBO or a *generative* latent variable model for $p(x)$, hence $p(x|z)$ is not modeled and no *decoder* is required. The GEDI objective instead includes an energy-based model $p_\psi(x)$.
> >     - The decoder reconstructs the data from latent variables and can be seen to prevent *collapse* in the latent space, since for two distinct images $x_i$, $x_j$ to be reconstructed, their latent variables $z_i$, $z_j$ must be distinct. So an *advantage* of the decoder is that it offers a principled way to avoid collapse, compared to the heuristic approaches of SSL methods.
> >
> > We hope we have answered all your queries and improved the paper. Let us know if we can address any additional questions or concerns and please consider updating your score if appropriate.

---

### Official Review · Reviewer_DybW · 2023-10-31

**Soundness:** 1 poor
**Presentation:** 2 fair
**Contribution:** 2 fair
**Rating:** 3
**Confidence:** 4

**Summary:**

This paper started with the motivation of a principled understanding of the latent processes of self-supervised learning and then argued that common SSL models learn representations that “collapse” in latent semantic clusters and lose the nuanced information such as style. To improve this, the authors presented SimVAE to enhance SSL. It is a hierarchical VAE by further factorizing the latent $p(z)$ into $p(z|y)p(y)$, where $y$ is the “semantic content” such as different classes (e.g., different dog breed classes) or different instances (different dog image samples). The choice of $p(y)$ is Gaussian or uniform, and $p(z|y)$ is a low variance Gaussian. The authors derived the ELBO bound for SimVAE and showed promising results on MNIST and FashionMNIST while also showing results not as competitive as other SSL methods on Celeb-A and CIFAR-10.

Despite good efforts, the current shape of the paper lacks many sound technical details to be accepted at ICLR.

**Strengths:**

Originality: using VAE for self-supervised learning is not particularly new [1-3], but the idea of building a VAE for SSL by considering semantic latent variables in a higher hierarchy and then using it to explain existing SSL algorithms seems new to the reviewer. The authors seem to design the method from first principles.

Quality: there are some promising empirical results on small datasets, such as MNIST and FashionMNIST, where the proposed method surpasses or reaches close to SSL methods such as SimCLR, VicReg, and MoCo. The derivation of the SimVAE method (Eqs. 1-8) is correct despite minor errata (the details are in the Questions section.)

Significance: bridging together generative and discriminative representation learning is an important topic, and the authors show their effort toward this step by trying to explain the underlying mechanisms of different SSL methods using a hierarchical VAE.

[1] Gatopoulos, Ioannis, and Jakub M. Tomczak. "Self-supervised variational auto-encoders." Entropy 23.6 (2021): 747.

[2] Zhu, Yizhe, et al. "S3vae: Self-supervised sequential vae for representation disentanglement and data generation." Proceedings of the IEEE/CVF Conference on Computer Vision and Pattern Recognition. 2020.

[3] Wu, Chengzhi, et al. "Generative-contrastive learning for self-supervised latent representations of 3d shapes from multi-modal euclidean input." arXiv preprint arXiv:2301.04612 (2023).

**Weaknesses:**

Originality: the authors did not discuss prior VAE SSL work, such as [1, 4].

Quality: this is the biggest weakness of this paper. Despite the good efforts shown by the authors, many important technical details are missing. The details are in the Questions section.

Clarity: coupled with the last point, reading certain parts of the draft can be challenging as some terms are not clearly defined or certain steps are missing. The details are also in the Questions section.


[1] Gatopoulos, Ioannis, and Jakub M. Tomczak. "Self-supervised variational auto-encoders." Entropy 23.6 (2021): 747.

[4] Nakamura, Hiroki, Masashi Okada, and Tadahiro Taniguchi. "Representation Uncertainty in Self-Supervised Learning as Variational Inference." Proceedings of the IEEE/CVF International Conference on Computer Vision. 2023.

**Questions:**

1. *Page 1, “but do not fit its posterior due to their discriminative nature”*
* It is unclear how to define “fit” and why due to the discriminative nature.

2. *Page 5, “and for any $z \in \mathcal{Z},$ a distribution $p(x|z)$ is implicity defined by the probabilities of samples mapping to it $\{ x \in \mathcal{X} | f(x) = z \}$”*
* It is unclear what it means to be “implicitly defined by the probabilities of samples mapping to it”. This is a vague statement without mathematical backing.

3. *Page 5, “Hence geometric properties of representations implicitly correspond to latent probabilistic assumptions.”*
* The authors could have shown theoretically rigorous proof to show this. And it is unclear what specific geometric properties the latent probabilistic assumption induced.

4. *Page 5, “And $z$ may be only identifiable up to certain symmetries”*
* The authors may specify what “symmetries” mean exactly in terms of the identifiability results and may cite related works.

5. *Page 5, “and insight into the information captured by representations from their regenerations”*
* The reviewer is not sure this claim is valid without further explanation; why do generative models have better insights into the information captured by the representation? It is better to define the “information” here, as in SSL literature, there are numerous works studying the information the representation captures (some of which the authors rightfully cited) [5-9].

6. *Page 5, “Note that if the variance of each $p(z|y)$ is low relative that of $p(z)$, this fits with the notion that contrastive methods ‘pull representations of related samples together and push those of random samples apart.’”*
* It would be much more valid if the authors showed proof of this. Also, it is not clear how to define rigorously “fits with the notion,” e.g., via asymptotic analysis. And how to quantify “low relative that of p(z)” is unclear.

7. *Page 5, “Thus, the model (Equation 4) justifies representation learning methods that heuristically perform clustering”.*
* It is unclear how it is justified. Factorizing $p(x)$ into the form of Equation 4 is a good start, but it did not justify why heuristic clustering methods are working well or necessarily capturing $p(x)$ well, e.g., through a tight error bound.

8. *Page 6, “samples of each class differ only in style (and classes are mutually exclusive) this collapse leads to style-invariant representations.”*
* Despite correct intuition, this statement is, in general, very strong; Dosovitskiy et al. did not explicitly claim anything about the style vs. semantic information in the representations, and the authors did not cite any other work supporting this claim nor specify any assumptions.

9. *Page 6, “Under softmax cross entropy loss for mutually distinct classes (cf mixed membership), all representations of a class $y$ converge to class parameter $w_y$.”*
* It is quite unclear what “representations converging to class parameter” means without any additional context. Also, the authors did not show any convergence analysis.

10. *Page 6, “In expectation, $z^T z’$ for stochastically sampled $z’$ of the same class approximates $z^T w_y$, without the need to store $w_y$.”*
* It is not mentioned at all why it $z^T z’$ approximates $z^T w_y$, and what “store $w_y$” means.

11. *Page 6, “In effect, representations are comparable to those learned by softmax, subject to unit length constraint.”: the authors may clarify how to define “comparable.”*
* It may be helpful to at least cite related work directly, or show empirical evidence to show under what tasks the representations are comparable.

12. Typos: Eq.(5) the support could be simply $y$ for the last integral, and in the paragraph below Eq.(5) the lower bound should be $\log p_{\theta}(z) \geq \int_{y} \mathbf{q_{\phi}(y|z)} \log \frac{p_{\theta}(z|y)p(y)}{q_{\phi}(y|z)}$ (the main result in Eq.(5) is correct).

[5] Tschannen, Michael, et al. "On mutual information maximization for representation learning." ICLR 2020,

[7] Wu, Mike, et al. "On mutual information in contrastive learning for visual representations." arXiv preprint arXiv:2005.13149 (2020).

[6] Sordoni, Alessandro, et al. "Decomposed mutual information estimation for contrastive representation learning." ICML 2021.

[8] Tsai, Yao-Hung Hubert, et al. "Self-supervised learning from a multi-view perspective." ICLR 2021.

[9] Mohamadi, Salman, Gianfranco Doretto, and Donald A. Adjeroh. "More synergy, less redundancy: Exploiting joint mutual information for self-supervised learning." ICIP 2023.

---

> ### Author Response · Authors · 2023-11-20
> **Rebuttal by Authors**
>
> * **"Originality: the authors did not discuss prior VAE SSL work."**: Thank you for raising these works. After carefull consideration, we cited [3] and [4] in the updated manuscript, in section 2 while we do not believe [1] is closely related to our work for the following reasons:
> [1] does not tackle the typical self-supervised learning problem we consider, of learning representations from semantically related data. Instead, they target richer and higher-quality generation using VAEs, by applying a "self-supervised" *lossy* augmentation to the data (e.g. downscaling or edge detection) that is assumed part of the generative process, allowing that process to be considered in stages (resembling a diffusion-like approach).
> [3] combines several loss components from SSL and VAEs to arrive at a comparable training objective to SimVAE's but does not propose a hierarchical latent variable model for SSL and does not derive its associate lower bound.
> [4] is related to our work but differs from SimVAE as it does not propose a generative latent variable model for SSL in the same way, rather their model $p(x|z)p(z)$ is defined in terms of a posterior and the data distribution itself (their Eq 9).
>
> * **Clarity & Quality**: we have reworded key sections of the paper to improve clarity and we address specific questions below.
>
> 1. **"but do not fit its posterior due to their discriminative nature"**: this has been rephrased as it is an unecessary level of detail for the introduction.
> 2. **"$p(x|z)$ is implicity defined by ..."** we have reworded this for clarity.
> 3. **"geometric properties of representations implicitly correspond to latent probabilistic assumptions"**: we have reworded this for clarity and more clearly defined the relationship between an encoder $f$ and the latent distribution it induces $p_f(z)$.
> 4. **"$z$ ... only identifiable up to certain symmetries"** this has been noted previously for generative models (c.f., Locatello et al. 2019), in particular VAEs, and we have added citations.
> 5. **"insight into the information captured by representations"**: we mean this less quantitatively than suggested and have amended the wording to make this clear.
> 6. **"variance of each $p(z|y)$ is low relative that of $p(z)$"**: at this point in the paper, we aim to be intuitive and motivate the steps we take. Note that we do not *prove* properties of the latent space learned by SSL methods, but draw intuition from the latent structure they impose, design a latent variable model based on that and aim to justify our choices empirically. We have rephrased "low relative to that of $p(z)$" as "more concentrated" and a concrete example of what we mean is if $p(z|y)$ and $p(z)$ are both Gaussian and the variance of former is lower than the latter.
> 7. **"the model (Equation 4) justifies representation learning methods that heuristically perform clustering"**: we have rephrased this section to hopefully make the explanation more clear. Maximising Eq 4 learns the model in Eq 3, and is a prinicipled means of training a VAE with mixture model prior. Methods that take a similar but heuristic approach can therefore be interpreted as also (approximately) learning the model in Eq 3.
> 8. **"samples of each class differ only in style (and classes are mutually exclusive) this collapse leads to style-invariant representations"**: we have reworded for clarity. We reference "Variational Classification" [Dhuliawala et al., 2023,] regarding the latent perspective and "collapse" of representations of each class under softmax. We agree that Dosovitskiy et al. do not reference content/style, we meant to refer to their reference to "transformation invariance" and have made this more clear.
> 9. **"representations of a class $y$ converge to class parameter"**: we have reviewed this wording and removed this part as it is unnecessary for the point we hope to make.
> 10. **" $z^T z’$ approximates $z^T w_y$, ... what “store $w_y$” means."** we have reviewed this wording and removed this part as it is unnecessary for the point we hope to make.
> 11. **"representations are comparable to those learned by softmax"**: we have re-worded to make this more clear. We aim to highlight common high-level clustering of representations of semantically related samples across various SSL methods.
> 13. typos: thank you, these have been fixed.
>
> We hope we have answered all your queries and improved the paper. Let us know if we can address any additional questions or concerns and please consider updating your score if appropriate.

---

> > ### Comment · Reviewer_DybW · 2023-12-03
> > **Response to the authors**
> >
> > The reviewer sincerely appreciates the response from the authors. Unfortunately, since the changes were not highlighted in the draft, it took the reviewer quite some time to verify them, and the reviewer was not able to validate them before the author-reviewer discussion period.
> >
> > Overall, the changes are positive, and the paper indeed has promising results. After reading the responses from other reviewers, the reviewer suggests that the authors further improve the most updated draft and submit it to the next venue.

---

### Official Review · Reviewer_2gEb · 2023-10-31

**Soundness:** 3 good
**Presentation:** 3 good
**Contribution:** 3 good
**Rating:** 6
**Confidence:** 4

**Summary:**

This article is placed in the context of representation learning using self-supervised learning (SSL) algorithms. It insists on the distinction between discriminative and generative SSL algorithms. The authors claim that while the former are generally easier to implement & train, and seem to generally produce better latent representation, they are actually very opinionated on which information is kept in the latent representation and which is discarded. The author argue that this is a result of the discriminative nature of the training process, which tends to only keep information necessary for the discriminative task and discard the rest. On the other hand, generative algorithms must retain as much information about the data as possible to fulfill their reconstruction training objective, and thus are theoretically capable of producing richer representation that contain more of the information from the data.

To try and bridge the empirical gap between those two families, the authors propose a graphical model representation that generalize the structure of many discriminative SSL algorithms, and use it to build a generative SSL model: SimVAE. It uses a hierarchical latent structure to encode the information that some training examples are related to each other without encouraging the model to discard information differing between them.

The proposed SimVAE is show to improve over other generative SSL models for downstream classification from their learned representation, in some cases being competitive with discriminative algorithms. Evidence is also given to the fact that SimVAE does learn richer representations than discriminative models, allowing better classification performance on secondary characteristics of the data.

**Strengths:**

This article seems rather solid. The proposed model is well motivated and theoretically sound.

The proposed latent construction for SimVAE is well adapted to problem of interest, and is and adequate answer to the claim that discriminative SSL tends to discard any information not relevant to the implicitly assumed class of downstream tasks.

The empirical evaluation of the proposed SimVAE is detailed, and performed against many relevant models. I am overall confident in the correctness of the results and relevance of the model.

**Weaknesses:**

**Observation model:**

As is unfortunately very common in the VAE literature, barely any discussion is done regarding the probabilistic model of the decoder, $p(x|z)$ (in this case, that would be the variance associated with the MSE loss). It has been shown that it controls the signal/noise trade-off of the model, and thus how much information is stored in the latent representation of VAEs, which is of particular interest here (see for example [Dosovitskiy and Brox, 2016](https://proceedings.neurips.cc/paper/2016/hash/371bce7dc83817b7893bcdeed13799b5-Abstract.html), [Rezende and Viola, 2018](https://arxiv.org/abs/1810.00597), [Loaiza-Ganem and Cunningham, 2019](https://proceedings.neurips.cc/paper/2019/hash/f82798ec8909d23e55679ee26bb26437-Abstract.html), [Berger and Sebag, 2020](https://arxiv.org/abs/2003.01972), or [Langley et al, 2022](https://arxiv.org/abs/2205.12533) for discussions about the observation model).

As a result, I believe that this parameter has potentially a large impact on SimVAE's performance as a representation learning method, and leaving it to $1.0$ (according to appendix A.4.3) is likely to be too large a value, causing the model to discard significantly more information than appropriate.

**Hierarchical VAEs:**

The idea of hierarchical VAEs built on a chain of latent variables is not new, and there is a wealth of models build on latent structures similar (if not identical) to SimVAE. While as far as I remember SimVAE is not redundant with these works, I find it lacking that they are not mentioned in the paper, and that SimVAE is not positioned relative to them. A few non-exhaustive examples: [Rolfe, 2016](https://arxiv.org/abs/1609.02200), [Dilokthanakul et al, 2017](https://arxiv.org/abs/1611.02648), [Edwards and Storkey, 2016](https://arxiv.org/abs/1606.02185), [Bouchacourt et al, 2018](https://ojs.aaai.org/index.php/AAAI/article/view/11867) or [He et al, 2019](https://openreview.net/forum?id=SJgsCjCqt7).

**Minor points:**

I think it would be an improvement to explicitly state what models of $p(y)$ and $p(z|y)$ are used in your experiments among the various possibilities that are suggested in Section 4, and how the training loss given in Algorithm 1 is derived from them.

**Questions:**

I don't have more questions beyond the points raised above.

---

> ### Author Response · Authors · 2023-11-20
> **Rebuttal by Authors**
>
> * **Impact of $p(x|z)$ variance** Thank you, we agree with the role this parameter has in governing the amount of information representations learn. We had considered this parameter in preliminary experiments but not fine tuned it for final experiments. We note that, for unsupervised representation learning, one should be cautious about tuning hyperparameters to a particular downstream task. However, following the reviewer's suggestion, we have re-run experiments for several values of $\sigma^2=Var[X|Z]$ ($\sigma^2 = 1.0$ in the paper) and include results in Figure 7 for downstream CIFAR10 classification. This shows a slight performance improvement ($+2\%$) from using $\sigma^2=0.6$. We are testing lower values to see if there is a better value "in general" (i.e. across various datasets) and if so will update the results. (We do not expect this change the overall picture, e.g. to "brige the gap" to discriminative performance.)
> * **Hierarchical VAE missing references**: Thank you for your feedback. We agree SimVAE fits within the wider hierarchical VAE literature and this was an omission. The related work (section 2) has been updated to include prior relevant hierarchical latent variable models ( Valpola, 2015; Ranganath et al., 2016; Rolfe, 2017; He et al., 2018; Sønderby et al., 2016; Edwards & Storkey, 2016) and how they relate to SimVAE. The closest work by Edwards & Storkey (2016), proposes a very similar hierarchical graphical model for modelling data*sets*. The main differerences to our work are (a) that we present a hierarchical latent variable to **unify existing SSL methods** (purpose); and (b) how the posterior is factorised, which is crucual for representations $z$ to be inferred for each sample $x$ independently of related $x'$ (note that Edward & Storkey first learn a global statistic $c$ from a dataset via $q(c|D)$ and only then infer a representation of the data $q(z|x,c)$).
> * **Model specifications**: We have clarified the distribution assumptions used in the  model in §5. Section A.3 has been reworded to tie the ELBO formulation to Algorithm 1.
>
> We hope we have answered all your queries and improved the paper. Let us know if we can address any additional questions or concerns and please consider updating your score if appropriate.

---

### Official Review · Reviewer_Tr95 · 2023-11-07

**Soundness:** 2 fair
**Presentation:** 3 good
**Contribution:** 2 fair
**Rating:** 5
**Confidence:** 5

**Summary:**

The authors propose a hierarchical latent variable model for self-supervised representation learning. The lower-level latent variables correspond to the learned representations while the higher-level latent variables correspond to class/clusters. The authors propose an ELBO to the marginal log-likelihood and propose an algorithm to optimize the ELBO. The authors demonstrate that the resultant representations outperform representations learned by VAE.

Other than that, the authors propose variational approaches for performing instance discrimination, deep clustering etc.

**Strengths:**

The primary strength of the model is that it follows from first principles.  The learned features are diverse and preserve stylistic information as compared to discriminative approaches.

**Weaknesses:**

There are several weaknesses in this paper:

1) The paper is very hard to read. The primary contribution of the paper is equation (7) defined over J semantically related samples. The rest of the paper is filled with a lot of claims that do not belong to the paper. For instance, section 4.1 has a latent variable approach to instance discrimination. It is neither interesting nor surprising that a latent-variable version of instance discrimination or any other model can be created. Unless it serves some purpose or offers extra insights, it should be removed.
Everything except 4.2 needs to be removed from section 4.

2) Having an entire section for representation learning is again wasteful. The representation learning section needs to be moved to related work.

3) The authors should include the algorithm in their main paper rather than keeping it in the appendix.

I have put other issues in the Questions section

**Questions:**

1) What is the purpose of adding equation 9) since J=6 is used during training and J=2 is never used?
2) Which equation is used during training? Which equation corresponds to Algorithm 1? If it is equation 8), what is q(y|z1, ..., zJ). Infact it is necessary to show how  each of the distribution is represented.

---

> ### Author Response · Authors · 2023-11-20
> **Rebuttal by Authors**
>
> * **"Authors propose variational approaches for performing instance discrimination, deep clustering"/"Equation 7 is the primary contribution of the paper"**
>     Thank you for your feedback. As described in Section 1, our contributions are two-fold:
>     * [Section 3 \& 4.1] we perform an analysis of existing discriminative SSL methods (i.e., contrastive learning, instance discrimination, latent clustering). By taking a latent variable perspective, we show how these methods can be tied back to a common framework.
>     * [Section 4.2] we leverage this analysis to formally define the graphical latent variable model (i.e., Figure 2) which unifies the aforementioned methods and for which we perform experimental validation to verify its soundness and how it can help improve downstream prediction over existing generative and SSL methods.
>
>     We do not propose novel variational approaches for instance discrimination or deep clustering, but rather a latent variable model to unify these methods. §3 & §4.1 formalise a relationship between discriminative and generative methods and define the latent variable model to motivate SimVAE.
>
>
> * **"Hard to read"**: Thank you for this feedback, which we have taken seriously and reworded the manuscript (in particular §1-4) to improve clarity. Additional care was given to the detailed derivation of the proposed objective as well as its connection to the computational steps performed in practice (c.f., section A.3).
>
> * **"Claims that do not belong to the paper"**: As suggested, part of the representation learning section (§3) has been moved to Background & Related Work (§2). Sections 3 & 4.1 have been reworded to make clearer their contribution and connection to the rest of the paper. §3 focuses on the relationship between generative and discriminative approaches to representation learning, makes this more formal than previously and is hopefully more clear. This shows that both methods can induce latent structure, hence we analyse various discriminative SSL methods (§4.1) to show that they induce comparable latent structure, then take a generative approach to achieve the same (§4.2). Without this link, we wouldn't provide a rationale for those existing SSL methods, we would simply be proposing a new approach.
>
> * **"Algorithm in the main paper"**: Agreed, we have reformatted the paper to accommodate this.
>
>
> * **"Purpose of equation 9"**
>     This was included for clarity since many SSL methods compare pairs of data ($J=2$). However, we agree this is not necessary and have removed it to the appendix.
>
> * **"Equation used during training"**
>     Thank you for your feedback. We reworded section 4.2, 5 and A.3 for greater clarity regarding the SimVAE objective, the involved distribution (including $q(y|z)$) and the connection between the ELBO and Algorithm 1, respectively.
>     * Equation 7 refers to the ELBO used to train SimVAE in the general case.
>     * In the paragraph below Eq. 7 we specify assumptions that we make (which for different data sets may be varied).
>     * In particular, we assume following:
>         * $q(z|x)$ and $p(x|z)$ are Gaussian, as in in typical VAE.
>         * $p(z|y) = \mathcal{N}(z; \mu_{y}, \sigma^2)$ are Gaussian, with (small) fixed common variance $\sigma^2$.
>         * that $y$ is continuous and uniformly distributed, allowing $y$ to be integrated out (as described in §4.2).
>
>     These assumptions are reflected in Algorithm 1, which we implement in the experiments.
>
> We hope we have answered all your queries and improved the paper. Let us know if we can address any additional questions or concerns and please consider updating your score if appropriate.

---

### Author Response · Authors · 2023-11-20
**General Response to All Reviewers**

We thank all reviewers for their time and consideration of our work and for aknowledging the **soundness** (Tr95, 2gEb, DybW), **relevance** (2gEb, DybW, xsXw), **novelty** (2gEb, DybW), and **benefits brought by SimVAE** (Tr95, 2gEb).

We address the following general concerns raised by the reviewers in our updated manuscript:
* **Presentation & Readability** (Tr95,DybW): Sections 2,3,4,5,6 have been reworded to improve readability.
* **Difficulty to tie the theoretical analysis with the experimental work** (Tr95, 2gEb): the following sections have been reworded for clarity:
    * §4.2: now mentions that Eq 7 refers to SimVAE's ELBO
    * §5: clarifies distributional assumptions of the model
    * A.3: derives Eq 7, describes distribution choices and links them to Algorithm 1.
* **Unsuported claims/Missing references** (2gEb,DybW,xsXw): Connections with hierarchical VAEs and missing SSL references have been added to §1,2. §3,4 have been updated to more clearly justify or reference all claims.

In addition, we highlight the addition of the following material to the results section.
* **Additional results**: In §6, we add Fig 3.b to the previous table for one Celeb-A attribute prediction task), which summarises SimVAE's benefit across _20 attribute prediction tasks_. Figure 3.b shows that, on average, SimVAE outperforms _all generative and discriminative baselines_ on these tasks, strengthening our claim that SimVAE captures more information in its representations.

---

### Meta-Review · Area_Chair_9FdT · 2023-12-08

**Metareview:**

The paper makes a fair contribution of a framework trying to unify (some) existing discriminative representation learning methods and which also elicits a generative approach, which demonstrates the benefit of the more comprehensive information in the generatively learned representation (e.g., better for "style-oriented" tasks). Nevertheless, reviewers overall tend to regard this is not yet done in satisfying quality. The interpretation of discriminative methods under the graphical model still seems a bit superficial and formal, and is weakly connected to the proposed method. It is also expected to have more discussion/comparison with other generative representation learning methods. Moreover, there are quite a few rephrasing and reorganizing edits and new results, for which more review efforts would be needed to verify the clarity, validity, and reproducibility. I hence recommend a reject for the submission in the current form and encourage the authors to further improve the quality accordingly.

**Justification For Why Not Higher Score:**

The mentioned weaknesses render the quality of the paper in current form insufficient for ICLR; please refer to the Metareview for more details. I posted this recommendation in the AC-reviewer discussion period, and no reviewer further argued for acceptance.

**Justification For Why Not Lower Score:**

N/A

---

### Decision · Program_Chairs · 2024-01-16

Reject